# Design-rules for stapled peptides with in vivo activity and their application to Mdm2/X antagonists

Arun Chandramohan[1], Hubert Josien[2], Tsz Ying Yuen[3], Ruchia Duggal[4], Diana Spiegelberg [5], Lin Yan[2], Yu-Chi Angela Juang[1], Lan Ge[2], Pietro G. Aronica [6], Hung Yi Kristal Kaan[1], Yee Hwee Lim[3], Andrea Peier[2], Brad Sherborne[2], Jerome Hochman[7], Songnian Lin[2], Kaustav Biswas[4], Marika Nestor [8], Chandra S. Verma [6], David P. Lane[9], Tomi K. Sawyer[4], Robert Garbaccio[2], Brian Henry [1]✉, Srinivasaraghavan Kannan [6]✉, Christopher J. Brown[9]✉, Charles W. Johannes [3,9,10]✉ & Anthony W. Partridge [1,11]✉

Although stapled α-helical peptides can address challenging targets, their advancement is impeded by poor understandings for making them cell permeable while avoiding off-target toxicities. By synthesizing >350 molecules, we present workflows for identifying stapled peptides against Mdm2(X) with in vivo activity and no off-target effects. Key insights include a clear correlation between lipophilicity and permeability, removal of positive charge to avoid off-target toxicities, judicious anionic residue placement to enhance solubility/behavior, optimization of C-terminal length/helicity to enhance potency, and optimization of staple type/number to avoid polypharmacology. Workflow application gives peptides with >292x improved cell proliferation potencies and no off-target cell proliferation effects ( > 3800x on-target index). Application of these 'design rules' to a distinct Mdm2(X) peptide series improves ( > 150x) cellular potencies and removes off-target toxicities. The outlined workflow should facilitate therapeutic impacts, especially for those targets such as Mdm2(X) that have hydrophobic interfaces and are targetable with a helical motif.

Only a small portion of the proteome is considered druggable with traditional approaches[1,2]. Specifically, monoclonal antibodies are restricted to the extracellular space and high-affinity engagement with small molecules is generally limited to deep hydrophobic pockets such as those found at enzymatic active sites. Thus, target classes such as transcription factors and intracellular protein-protein interactions (PPIs), remain largely out-of-reach. Stapled α-helical peptides (stapled peptides) represent a potential corresponding solution[3]. Indeed, these

[1]MSD International, Singapore 138665, Singapore. [2]Merck & Co., Inc., Kenilworth, NJ 07033, USA. [3]Institute of Sustainability for Chemicals, Energy and Environment, Agency for Science, Technology and Research (ASTAR), Singapore 138665, Singapore. [4]Merck & Co., Inc., Boston, MA 02115, USA. [5]Department of Surgical Sciences, Uppsala University, Uppsala, Sweden. [6]Bioinformatics Institute, Agency for Science, Technology and Research (ASTAR), Singapore 138671, Singapore. [7]Merck & Co., Inc., West Point, PA 19486, USA. [8]Department of Immunology, Genetics and Pathology, Uppsala University, Uppsala, Sweden. [9]Institute of Molecular and Cell Biology, Singapore 138673, Singapore. [10]EPOC Scientific LLC, Stoneham, MA 02180, USA. [11]Present address: Genentech, South San Francisco, CA 94080, USA. ✉e-mail: brian.henry3@msd.com; raghavk@bii.a-star.edu.sg; CJBrown@imcb.a-star.edu.sg; cwjohannes@gmail.com; partridge.anthony@gene.com

molecules can bind with high affinity to these targets' flat surfaces to result in modulation of protein function[4]. Stapled peptides contain one or more specialized macrocycles that promote the α-helical conformation. Olefin staples, the most common staple type, typically incorporate two α-methylated amino acids at either i –> i + 4 or i –> i + 7 positions, linked via ring closing metathesis (RCM)[4,5]. Drug-like properties result, including enhanced binding, stabilized helical structure, enhanced proteolytic stability, and in some cases, cell permeability[6]. However, achieving sufficient cell permeability to engage the target while avoiding off-target effects has proven to be a central challenge. Accordingly, we sought to uncover stapled peptide 'design rules' by studying an existing class of cell-active peptides that disrupt the p53/Mdm2(X) interaction.

p53, the "guardian of the genome" is a transcription factor that regulates a variety of anti-oncogenic processes such as DNA damage repair, cell cycle arrest, apoptosis, and prevention of angiogenesis[7]. The importance of these functions is cemented by the observation that p53 is the most mutated protein in human cancers[8]. As well, Mdm2 and MdmX (also known as Mdm4), critical negative regulators of p53 function, are commonly upregulated in a variety of human lesions[9]. Both proteins inhibit p53 function through a sequestrating interaction involving the p53 N-terminal α-helix. Mdm2, but not MdmX, also acts as an E3 ligase to further suppress p53 activity by tagging it with polyubiquitin, leading to p53 destruction in the proteasome[10].

The p53/Mdm2 PPI represents a valuable opportunity to make cross-modality comparisons since both small molecule and stapled peptide antagonists have been developed[7]. Stapled peptides consisting of either L- or D-amino acids have been discovered with both classes of molecules binding to Mdm2 and MdmX to displace the p53 N-terminal helix[6,11–14]. The L-peptides include the clinical compound ALRN-6924 and its well-characterized progenitor ATSP-7041[6,13,15]. Several small molecule inhibitors have also advanced to the clinic[16]. These bind to the same Mdm2 pocket as the stapled peptides but lack the capacity to strongly bind to and inhibit MdmX, thus the Mdm2/X dual-inhibitory nature typical of stapled peptides such as ATSP-7041 may represent a differentiating advantage[13]. Understanding this and how the drug-like properties of these two distinct modalities impact therapeutic potential could play an important role in strategic drug discovery investments.

Despite their promise, the discovery of on-target, cellularly active stapled peptides has been challenging. Specifically, compounds that appear to show cellular activity are frequently confounded by off-target toxicities, most notably the induction of membrane lysis which can lead to false positives[17] and pose safety liabilities[18–20].

Here, we gain key understandings for the design-rules for a well-behaved, cell permeable/active stapled peptide. Specifically, we synthesize a library consisting of >350 individual ATSP-7041 analogs and test these for target binding, cellular activity, cellular permeability, membrane disruption, and counter-screen activity. We reveal insights including a correlation between lipophilicity and cell permeability, a tendency for cationic residues to cause off-target toxicities, while anionic residues are well-tolerated. Optimization of staple type and peptide length/sequence composition result in peptides with improved cellular activity, decreased off-target effects, and improved in vivo activities. Combining favorable attributes leads to a lead peptide with low-nanomolar cell proliferation inhibitory activity with no off-target cell proliferation effects. Application of these lessons culminates in a distinct Mdm2 antagonist stapled peptide series with no observable off-target toxicities and a cellular activity that is improved by >150-fold.

## Results

### ATSP-7041 shows on-target activity but with a narrow on-target window
Therapeutic advancement of stapled peptides are often confounded by cellular toxicity[17]. We leveraged ATSP-7041 and analogs (such as the previously studied azido-version, MP-081[6], Fig. 1a, b), as these had been previously validated to be high-affinity Mdm2(X) binders with bona fide cellular activity[6,21]. However, these peptides did not represent fully optimized stapled peptides as they exhibited off-target effects in multi-day cell proliferation studies (Fig. 1c). Specifically, although these peptides showed measurable proliferation potencies against cell lines harboring wild-type (WT) p53, they had a narrow on-target index, with activities against control lines (p53 null, p53 mutant or p53-depleted through HPV infection) that were only ~10-fold weaker (Fig. 1c). This poly-pharmacology was not observed with the cell impermeable control peptide (MP-950, Fig. 1c) but was seen with a cell permeable non-binding control peptide (MP-202, ATSP-7041 with a F3 to D-Phe substitution, Fig. 1c), confirming the effects to be independent from Mdm2/X binding. These results suggested that further study of this peptide series might provide insights for optimizing the cellular activity while removing the off-target toxicities.

### Lipophilicity correlates with cell permeability
To gain insights as to how to design well-behaved, cell-active stapled peptides against an intracellular target, we made >350 ATSP-7041 analogs. These represented single and multiple amino acid substitutions and were characterized with a spectrum of metrics including binding, cellular activity, activity in a p53 independent counter-screen, circular dichroism spectroscopy, experimental LogD, calculated ALogP, solubility, NanoClick permeability[22], and membrane disruption (as measured by lactate dehydrogenase (LDH) cytoplasmic release), (Supplementary Data 1). As the NanoClick cell permeability assay relies on intracellular click-chemistry, most peptides used the previously-studied[6] azide-ATSP-7041 (MP-081) as a template (Fig. 1a, b). Other library members were based on the parent peptide, ATSP-7041[13] or PM2[12], a related sequence (Fig. 1b).

Peptides included those with single or multiple amino acid substitutions at various positions. To leverage cell activity as an important metric, we generally preserved amino acid residues critical for the Mdm2 binding interface (F3, W7, Cba10), as defined in Fig. 1b (Cba corresponds to cyclobutyl alanine, a non-natural leucine analog). To avoid having time and serum binding components confound permeability measurements, we used 0% serum and a 4-hour peptide incubation time as our standard assay condition. 16-hour time-points and 10% serum conditions were also collected for a subset of peptides (Supplementary Data 1). The library represented peptides with diverse properties as shown by a wide range of values in biochemical and cellular assays as well as calculated attributes (Fig. S1). Significantly, this structurally-diverse ATSP-7041 library represents a valuable public resource for a collective understanding of α-helical peptide design. Our initial analysis focused on variants of azide-ATSP-7041 (MP-081, Fig. 1a, b) and led to the following key observations:

1. As expected, mutations to the three key binding residues (F3, W7, Cba10) generally had large decreases in binding affinity to shift the $K_D$ by >50x (Fig. 2a and Supplementary Table 1, parent $K_D$ = 5 nM). The only exceptions were substitution of W7 with tryptophan analogues (within 2x of the parent affinity) and substitution of Cba with Leu ($K_D$ = 56 nM) or Ala ($K_D$ = 115 nM). Outside of the Mdm2 binding motif, ~70% (47 out of 66) of the single substitutions maintained low nanomolar-affinity binding (< 50 nM) and all but one maintained affinities of <250 nM (Fig. 2a, Supplementary Table 1). The exception to the latter was E5P ($K_D$ = 7.8 μM), (Fig. 2a, Supplementary Table 1), attributable to proline-induced helical structure loss as circular dichroism spectroscopy measured helicity of this peptide at 22%, compared to 50% for MP-081 (Supplementary Data 1). A preponderance (8 out of 10) of substitutions at Thr2 led to weaker binding (Fig. 2a), which is consistent with a known structural role of the threonine sidechain at this position[6]. In addition, peptides with

**a**

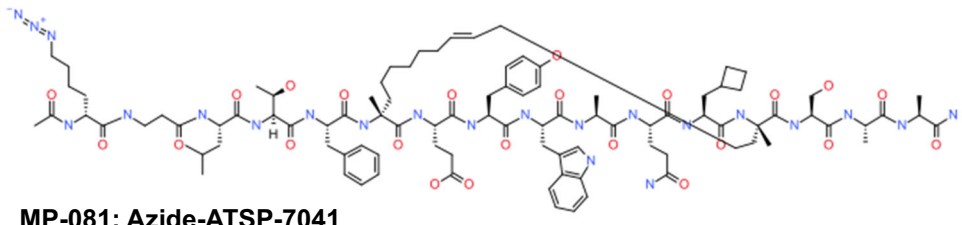

**MP-081: Azide-ATSP-7041**

**b**

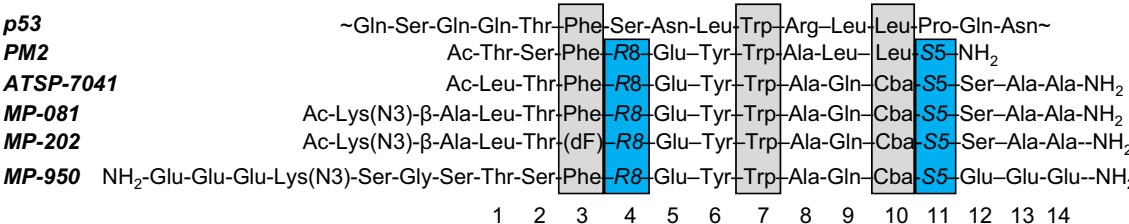

**c**

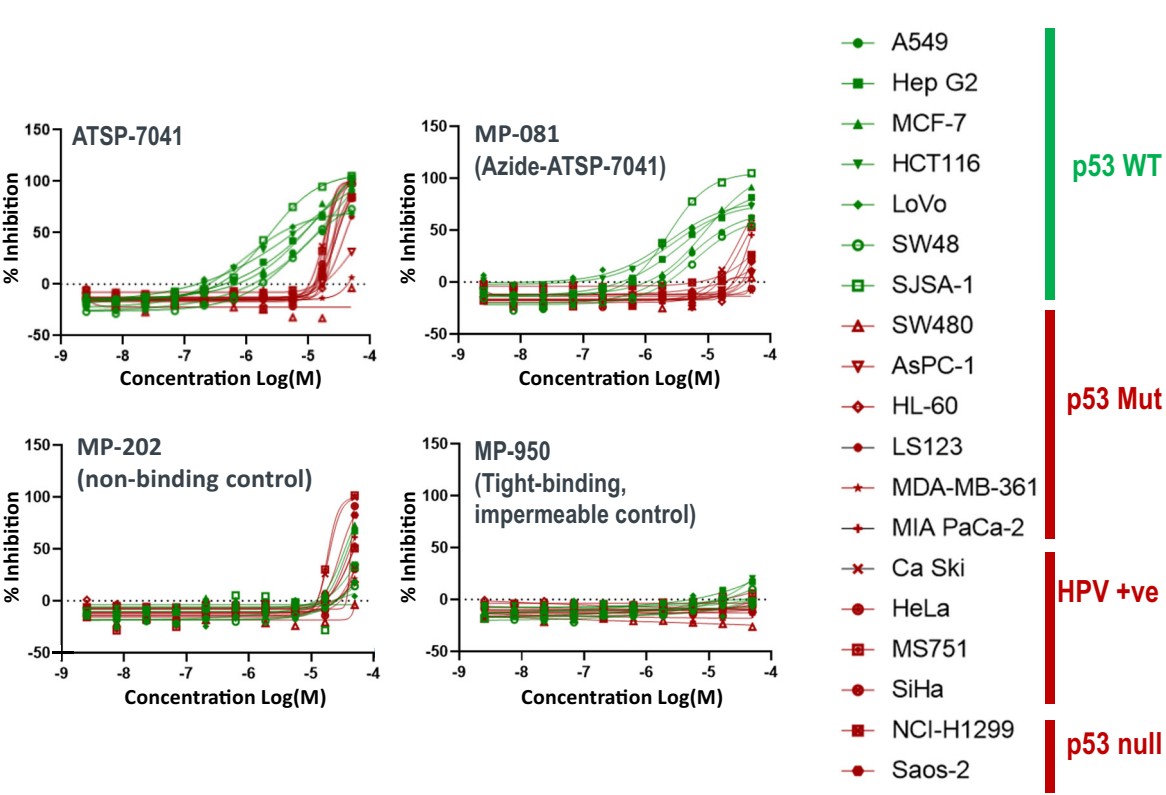

Fig. 1 | **Prototypic Mdm2/X dual antagonist stapled peptides show on-target cellular activity but with a narrow therapeutic index. a** Structure of azide-ATSP-7041 (MP-081), **b** Peptide sequences for the p53 N-terminus, library template molecules (PM2, ATSP-7041, MP-081, a non-binding control peptide (MP-202), and a high-affinity but impermeable control (MP-950). Residues highlighted in grey correspond to the critical binding positions whereas the residues highlighted in blue are the stapling positions. R8 and S5 refer to (R)−2-(7′-octenyl)alanine and (S)−2-(4′-pentenyl)alanine (respectively) used in the ring closing metathesis reaction to form the olefin staple. **c** Proliferation profiles of ATSP-7041, its azido-analog (azide-ATSP-7041, MP-081), and the non-binding (MP-202) and cell impermeant (MP-950) controls against a panel of cancer cell lines that are either p53 WT (green) or p53 deficient (i.e., p53 null, mutant, or p53-depleted through HPV infection, red). Compounds were tested in duplicate.

**a**　　　Effect of amino acid substitution on binding $K_D$ compared to MP-081

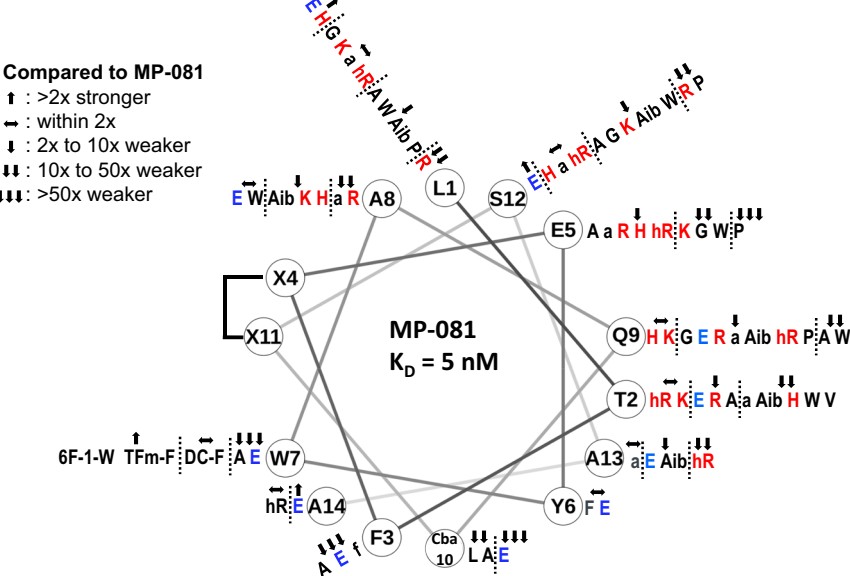

**b**　　　Effect of amino acid substitution on p53 reporter cell activity compared to MP-081

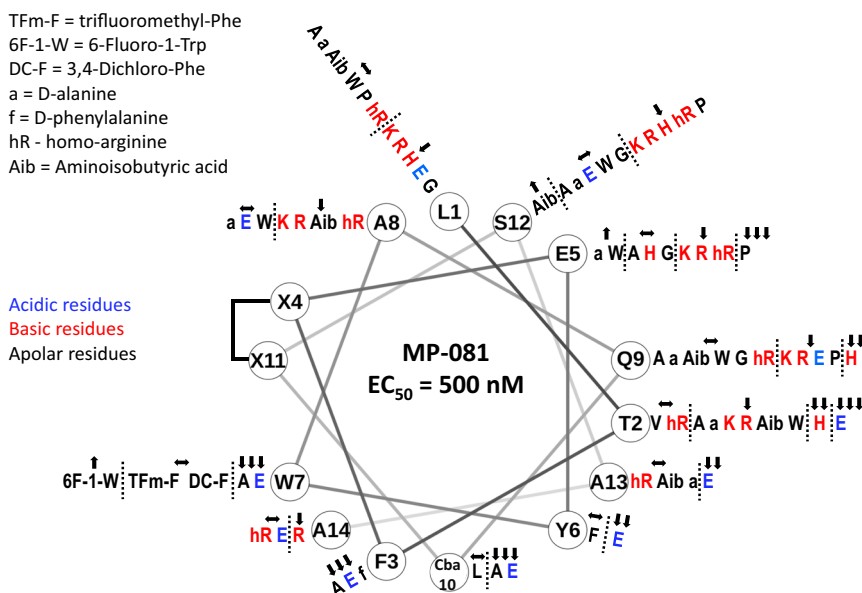

**Fig. 2 | A library of ATSP-7041 analogs uncovers design trends.** For clarity, the N-terminal azido-lysine and beta-alanine residues are not shown in the helical wheel diagrams. Compounds coloured in blue represent acidic residues, those in red represent basic residues, and those in black represent apolar residues. Compounds were tested in duplicate. **a** Aside from the critical contact residues (F3, W7, & Cba10), substitutions with a variety of amino acids tend to maintain high affinity binding (< 250 nM). **b** Single substitutions to charged residues (K, R, hR (homo-Arg), H, E) at non-binding positions, tend to result in decreased cellular activity while substitutions to other types of residues had varying affects.

poor solubilities tended to have shifted binding constants (Supplementary Table 1), values that are likely artifactual due to the molecules being sequestered into soluble aggregates. Overall, these results indicate that if the binding motif and staple are held constant, there is a large sequence space that could be explored to optimize other peptide properties while maintaining high affinity. Indeed, exemplary peptides with multi-site substitutions to a single amino acid had $K_D$s that were within 3-fold of the parent peptide; for example, azide-ATSP-7041 (L1D, E5D, A8D) and azide-ATSP-7041 (L1N, E5N, A8N, S12N, A13N, A14N), had $K_D$ values of 5 and 15 nM, respectively (Supplementary Table 1, MP-495 and MP-359, respectively).

2. At non-binding positions, single substitutions to charged residues (K, R, hR (homo-Arg), H, E) usually resulted in decreased cellular activity (Fig. 2b, 22 of 31 substitutions), an observation that was slightly more common with basic substitutions (20 of 26, 77%) versus substitutions to acidic residues (5 of 8, 63%). We noted that single substitutions to basic residues almost universally led to poor solubility (≤ 10 μM, Supplementary Table 1), an observation likely due to the generation of peptides with a net charge of zero (except for substitutions at the E5 position). Furthermore, substitution to histidine often led to counterscreen efficacies that were greater than the tetracycline control, with the T2H peptide as the worst offender (277% efficacy at 50 μM,

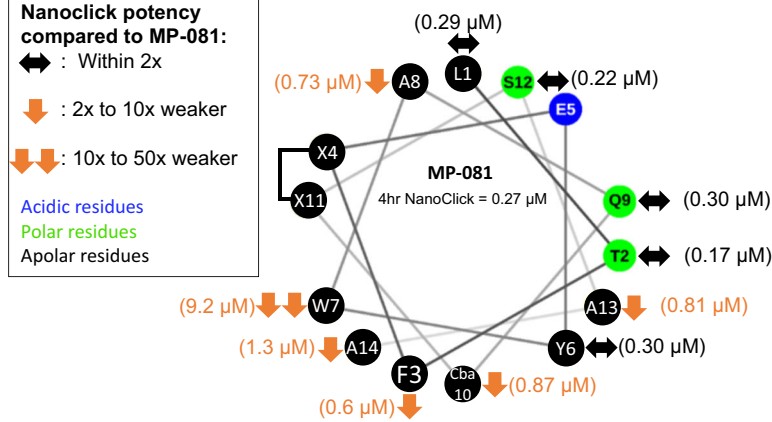

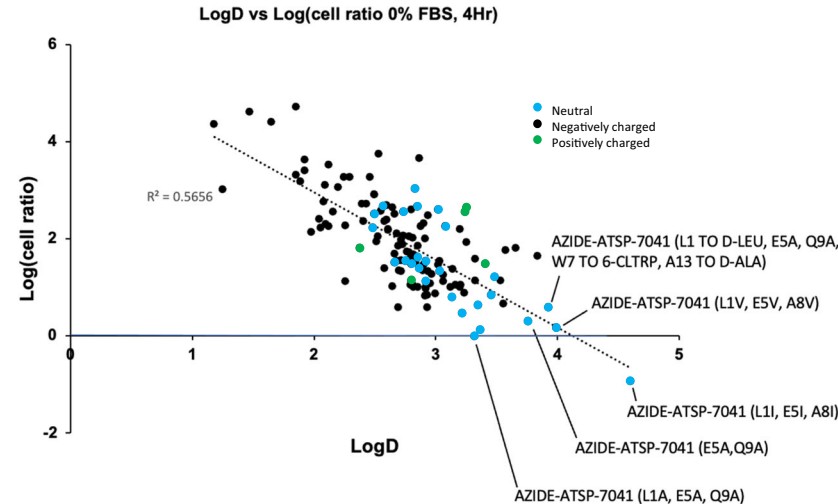

**Fig. 3 | Membrane permeability correlates with peptide lipophilicity, but acidic residues can be accommodated at select positions. a** The ATSP-7041 core sequence is amphipathic as demonstrated by a helical wheel depiction where residues are coloured according to polarity; residues in black are apolar, those in blue are acidic, and those in green are uncharged polar. Application of a full glutamic acid scan of this amphipathic sequence demonstrates that placement of a Glu residue on the apolar helical face tended to decrease NanoClick permeability whereas placement on the polar face was generally tolerated. **b** Library analysis reveals that permeability (cell ratio = cellular $EC_{50}$ / binding $K_D$) is correlated with lipophilicity (LogD). To avoid the confounding effects of polypharmacology, peptides included were restricted to those that had a Mdm2 $K_D$ value < 10uM and had cell activities that were at least 10x greater than the potencies in the counterscreen and LDH assays.

$EC50 = 5.9$ μM, Supplementary Table 2). As well, peptides containing multiple positive charges (e.g. MP-537, MP-538, MP-539, MP-879, MP-880) frequently resulted in LDH release and/or counterscreen activity (Supplementary Table 1). Further, there have been several literature reports[18–20] that highlight that the combination of hydrophobic character and positive charge can result in mast cell degranulation (MCD), a potentially fatal condition[18–20]. Overall, the results indicate that caution must be taken when introducing cationic residues.

3. Conversely, a full glutamic acid scan of MP-081 generated peptides with universally excellent solubility (>90 μM, Supplementary Table 1). Although introduction of Glu at most positions led to decreased cellular activity, peptides with an A8E, S12E, or A14E substitution had cellular potencies that were equipotent (within 2x) to the parent peptide (Fig. 2b). Importantly, none of the peptides with additional acidic residues showed off-target effects, even when peptides contained two (e.g., the peptides in Fig. 3a), three (MP-496), four (MP-3684) or even up to seven (MP-

950) glutamic acid residues, although these peptides were generally cellularly inactive (Supplementary Table 1). The Nano-Click assay[22] determines permeability of azide-labeled peptides by combining intracellular copper-free click chemistry with Nano-BRET technology. Specifically, if an azido-peptide achieves cytosolic exposure, it will react with a cyclooctyne-decorated Halotag construct that is fused with nanoluciferase. This reaction blocks the click labelling of a subsequently added azido-labeled acceptor dye, thus decreasing the NanoBRET signal in a manner that is target-binding agnostic. By applying this assay, we were able to study the permeability effects of the Glu scan, even at positions where target binding was abolished. We noted that the parent ATSP-7041 sequence is amphipathic, possessing a target-binding hydrophobic face (the hydrophobic staple and the apolar residues colored in black in Fig. 3a) and a solvent-exposed hydrophilic face (residues colored in blue or green, Fig. 3a). Introduction of a Glu on the hydrophilic side tended to maintain NanoClick permeabilities. In contrast, placing a Glu residue in the

hydrophobic face tended to reduce NanoClick permeabilities, with the W7E peptide showing the most dramatic effects (~34x shift). Overall, these results suggest that charge and polar separation can be incorporated into a-helical stapled peptide design to balance the dual needs for permeability and solubility.

4. A weak correlation between peptide helicity and cellular permeability was observed. Specifically, when permeability (as measured by the cell ratio, defined as the cellular $EC_{50}$ / Mdm2 $K_D$), was plotted versus peptide helicity, we noted that peptides with poor cell ratios (400 or greater) also had low helicity values (<31%, Fig. S2A). This observation suggests enhancing helicity may contribute to improved cellular permeability, as has been suggested by a previous study[23].

5. A stronger correlation was observed between peptide polarity and permeability. Indeed, polar to apolar substitutions tended to enhance cellular activity in our standard assay (0% serum, 4 hours). When we analyzed the entire library of ATSP-7041 analogs, we found that peptide lipophilicity (experimental LogD) correlated with both permeability (as measured by the cell ratio, Fig. 3b or Nanoclick, Fig. S3B) and cellular activity (Fig. S2C). The most potent peptides were those rendered net-neutral charge (Fig. 2d, peptides highlighted in blue) through a substitution at residue E5 to an apolar residue (Fig. 3b) and represented a significant increase in cellular potency, up to 10x (e.g., 0.044 μM for MP-965, AZIDE-ATSP-7041 (E5A, S12A), Supplementary Table 3). However, we judged that designing peptides which were increasingly hydrophobic (LogD > 3) was not a practical path to the clinic as these molecules also displayed poor solubilities (often <1 μM, Supplementary Table 3). In addition, peptides with either measurable poor solubilities (<10 μM) or those with poor behavior in our solubility assay (i.e., gave multiple peaks), showed >10x loss in cellular potency when tested in the presence of 10% FBS (Supplementary Table 3). Furthermore, we tested one such highly hydrophobic peptide (MP-350, LogD = 3.37) in a cell proliferation panel and it showed off-target proliferation effects in control lines that overlapped with the on-target effects (Fig. S3). Although increasing hydrophobicity may seem an attractive option at the in vitro stage, poor solution behavior would pose a considerable challenge during formulation and testing in vivo.

Next, we sought to improve solubility and reduce the observed off-target effects (Figs. 1c and S3). In addition to aiding in cell permeability (Fig S2A), we speculated that enhancing helicity would improve solubility by preventing β-sheet based peptide aggregation. Enhanced helicity might also limit alternative peptide conformations to mitigate toxicities related to promiscuous binding to off-target proteins. Accordingly, we sought to stabilize peptide helicity by I) extending the peptide with sequences of high helical propensity and II) optimizing both the number and nature of the peptide staples. To assess these effects, we shifted our primary focus to cellular activity in the presence of serum and measured at a later time point (16 hours) as it represented a more physiological condition.

## C-termini with high helical propensity enhance cellular activity

Inspection of the patent literature suggested that increasing the number of residues on the C-terminus of the peptide leads to improved solubility[24]. Specifically, Ala-rich C-terminal extensions (referred to here as C-terminal tails) were reported to increase peptide solubility. The most common patent sequences C-terminal to the staple were Ala-Ala-Ala-Ala-Ala-Ala-NH$_2$ ($A_6$-NH$_2$) and Ala-Ala-Ala-Ala-Ala-(D-Ala)-NH$_2$ ($A_5$-dA-NH$_2$). In our hands, when we replaced the MP-081's SAA-NH$_2$ tail with the $A_5$-dA-NH$_2$ tail (MP-464), we observed a modest improvement in solubility (Table 1), and cellular activity was improved

by ~3x (290 vs 880 nM, 10% serum, 16-hour assay, Table 1). A similar improvement in cellular activity was observed with the $A_6$-NH$_2$ tail (MP-464, Table 1), although this peptide did not behave well in our solubility assay. Interestingly, tails with six Ala residues appeared to be a 'sweet spot' as peptides containing a $A_5$-NH$_2$ (MP-685) or an $A_7$-NH$_2$ (MP-470 and MP-471) motif had slightly worse cellular activities in the presence of serum (Table 1). Notably, compared to the 6-residue tail peptides, those with additional Ala extensions ($A_7$-NH$_2$, MP-470 and MP-471; $A_9$-NH$_2$ motif, MP-759) had better cell potency (as low as 0.06 μM) in the 0%, 4-hour assay but had greatly right-shifted (up to 40x for MP-470) cellular potency in the 10% serum, 16-hour assay, presumably due to serum binding. Interestingly, when we studied the proteolytic stability of a subset of peptides in a protease-rich matrix (whole cell homogenate), we observed that both the $A_5$-dA-NH$_2$ and $A_6$.NH$_2$ peptides had short half-lives ($t_{1/2}$ values of 30 and 92 min for MP-032 ($A_6$-NH$_2$ tail) and MP-464 ($A_5$-dA-NH$_2$ tail), respectively), values that compare poorly to the parent peptide ($t_{1/2}$ > 24 hours). Metabolic profiling identified the A14/A15 peptide bond to be the proteolytic site, liberating an active metabolite with an additional negative charge at the C-terminus. Interestingly, a peptide representing a pre-cleaved tail (Ala$_3$-COOH, MP-684) had a cellular potency that was right-shifted by >5x in the presence of serum (Table 1), suggesting that the polyalanine tails are cleaved intracellularly. Indeed, this idea was supported by the fact that MP-464 ($A_5$-dA-NH$_2$ tail) was highly stable in serum plasma (Fig S4). We speculate that the intracellular generation of an uncapped C-terminus and its associated negative charge results in the peptide being better retained in the intracellular space, thus potentially contributing to the enhanced cellular activity. During the preparation of this manuscript, the sequence of the clinical molecule, ALRN-6924 (Sulanemadlin, MP-4897), was disclosed as Ac-LTF(R8)EYWAQL(S5)AAAAA(dA)-NH$_2$ (staple between R8 and S5)[25]. A detailed account of the discovery and characterization of this molecule was subsequently published[15]. We prepared this molecule and found that it behaved similarly to our azido-analogs (MP-032 and MP-464, Table 1). We also prepared a peptide similar to ALRN-6924 but with the (Ala)$_6$-NH$_2$ tail (MP-688) and found that it gave an equivalent potency in the 0% serum assay but a right-shifted potency in the 10% serum assay (1.3 μM, Table 1). The reason for the disconnect in potency between this peptide and our azido analog (MP-032) is not clear.

We hypothesized that the enhanced cellular activities of peptides with polyAla tails were related to the helical propensity of these segments. It is important to remember that peptides with diverse sequences—including those containing residues of low helical propensity—usually adopt a very high degree of helicity in low dielectric constant environments[26], such as the lipid bilayer and the membrane/water interface. Thus, stabilizing helicity in the aqueous phase might enhance permeability by decreasing the entropic penalty related to the transition to a highly helical membrane-inserted state. To challenge this hypothesis, we made a series of peptides with C-terminal tails that had either high or low predicted helical propensities in the aqueous environment (Table 1). Compared to MP-032 (-$A_6$-NH$_2$tail), all peptides containing tails with low helical propensity showed dramatic decreases in cellular activity (6.5x to >172x), despite maintaining high affinity for Mdm2. Except for the (AA-dA)$_2$-NH2 tail peptide, all molecules containing tails with low predicted helical propensity, had poor experimental aqueous helicity, as determined by circular dichroism spectroscopy (Table 1). In contrast, sequences containing tails with high helical propensity had cellular potencies similar to those observed with the polyalanine tail peptides. However, we noted that although these peptides generally had higher solution-based helicities compared to the low propensity group, experimental aqueous helicity values were not enhanced compared to MP-081. Thus, further studies will be required to fully understand how these tails lead to improved cellular activities.

**Table 1 | Azide-ATSP-7041 analogs with different C-terminal tails**

| Peptide | C-terminus (1st position is residue 14) | Series class | Binding $K_D$ (nM) | Cell $EC_{50}$ (4 hr, 0% FBS) (µM) | Cell $EC_{50}$ (16 hr, 10% FBS) (µM) | Predicted tail helical propensity (aqueous) | Aqueous solution helicity (%) | Solubility (µM) |
|---|---|---|---|---|---|---|---|---|
| MP-081 | -SAA-NH$_2$ | Template | 5.0 | 0.47 | 0.88 | High | 48.35 | 105.3 |
| MP-685 | -A$_5$-NH$_2$ | PolyA | 8.2 | 0.11 | 0.32 | High | 28.3 | N.D. |
| MP-032 | -A$_6$-NH$_2$ | PolyA | 5.0 | 0.15 | 0.29 | High | 34.6 | P.B. |
| MP-464 | -A$_5$-dA-NH$_2$ | PolyA | 31 | 0.49 | 0.29 | High | 32 | 157 |
| MP-470 | -A$_7$-NH$_2$ (Early isomer) | PolyA | 3.3 | 0.06 | 0.53 | High | 44 | N.D. |
| MP-471 | -A$_7$-NH$_2$ (Late isomer) | PolyA | 6.7 | 0.10 | 0.60 | High | 40 | N.D |
| MP-759 | -A$_9$-NH$_2$ | PolyA | 6.1 | 0.10 | 4.2 | High | 27.55 | 15 |
| MP-684 | -AAA-COOH | Cleaved tail | 11.5 | 0.25 | 1.7 | High | 32.6 | N.D. |
| MP-688* | -A$_6$-NH$_2$ | PolyA | 30 | 0.16 | 1.3 | High | 49.8 | N.D. |
| ALRN-6924* | -A$_5$-dA-NH$_2$ | PolyA | 2.5 | 0.14 | 0.32 | High | 51.25 | P.B. |
| MP-413 | -A-(Aib)$_5$-NH$_2$ | Helical tail | 1.7 | 0.45 | 0.71 | High | N.D. | N.D. |
| MP-966 | -AQA$_3$-dA-NH$_2$ | Helical tail | 45.9 | 0.39 | 1.0 | High | N.D. | 160 |
| MP-425 | -(AA-dA)$_2$-NH$_2$ | Non-helical tail | 4.2 | 1.0 | 2.6 | Low | 42.3 | 185 |
| MP-056 | -(A-dA)$_3$- NH$_2$ | Non-helical tail | 1.5 | 2.7 | 6.0 | Low | 22.23 | 157 |
| MP-443 | -A-Sar$_5$- NH$_2$ | Non-helical tail | 1.5 | 5.4 | 10 | Low | 30.45 | 180 |
| MP-289 | -G$_6$-NH$_2$ | Non-helical tail | 9.0 | 3.6 | 16 | Low | 24.33 | 142 |
| MP-093 | -P$_6$-NH$_2$ (Early isomer) | Non-helical tail | 132 | 15 | >50 | Low | 16.5 | 156 |
| MP-094 | -P$_6$-NH$_2$ (Late isomer) | Non-helical tail | 3.9 | 5.9 | 1.9 | Low | 18.7 | 158 |
| MP-896 | -dP$_6$-NH$_2$ | Non-helical tail | 11 | >50 | >50 | Low | 19.15 | P.B. |

Constant N-terminal sequence of Ac-K(N3)-βA-LTF(R8)EYWAQ(Cba)(S5)-, except for peptides marked with an asterisk which instead have an N-terminal sequence of Ac-LTF(R8)EYWAQL(S5)-. N.D.: Data not collected, P.B.: Poorly behaved; result inconclusive. Compounds were tested in duplicate.

## A variety of C-terminal tails give sub-micromolar cellular potencies

We sought to further enhance cell potency while avoiding off-target effects by optimizing the C-terminal tail sequence. Our multi-pronged approach involved optimization strategies, including: 1) solubility and solution state behavior through enhanced amphipathicity to balance the hydrophobic character; 2) enhancement of Mdm2 binding through productive interactions via the C-terminal tail, 3) tail stability through sequence optimization, residue a-methylation (addition of a methyl group to the amino acid's alpha-carbon, a modification that is known to inhibit protease cleavage[27]), and/or additional staples; and 4) increased cellular permeability, through enhanced interactions with the membrane.

To improve peptide behavior, we probed for amino acid positions that would tolerate polar residues without compromising cellular activity. Accordingly, we scanned the C-terminal tail (positions 12 to 17), of MP-464 (-A$_5$-dA-NH$_2$ tail) with substitutions of glutamic acid, a residue with high helical propensity (Table 2). Like the Glu-scan on the shorter MP-081 peptide, introduction of Glu residues generally decreased cellular activity, this included the A12E substitution (MP-834), which contrasts with what we observed with the shorter peptide. Interestingly, placement of a gamma-carboxyglutamic acid residue (Gla, a residue containing two acidic groups in the side-chain) at position 12 (MP-897), gave cellular activities that were within 2-fold of the parent peptide (Table 2). Similarly, a peptide with an A14E substitution (MP-833) gave a slightly enhanced cellular potency in the presence of serum (0.19 µM). In addition, a peptide with three additional Glu residues (MP-3685, AZIDE-ATSP-7041 (A8E, EEAAAA TAIL), Supplementary Data 1) had good cell reporter activity under serum

conditions (0.66 µM, Supplementary Data 1) and a strong HCT116 cell proliferation potency (0.64 µM, Supplementary Data 1), despite its overall -4 charge. These latter observations demonstrate that multiple negative charges can be accommodated in a cell-permeable stapled peptide. Most importantly, although peptides with the original A$_6$-NH$_2$ or A$_5$-dA-NH$_2$ tails tended to show some off-target proliferation effects in a counterscreen line that is devoid of cellular p53 protein (Ca Ski, Table 2), peptides with the additional Glu residues were largely devoid of proliferation effects in this line (Table 2). In addition, many of the peptides with Glu-containing tails had HCT-116 proliferation effects that approximated—or improved on—those of MP-464 or ARLN-6924. Overall, although the extra negative charge did not appear to markedly enhance peptide equilibrium solubility compared to MP-464 (Table 2), the increased anionic nature of these peptides may be assisting with peptide solution behavior in more subtle ways (i.e., solution micro-aggregation (oligomerization) state and/or dissolution kinetics) to result in potent molecules with very good on-target profiles.

Examination of the helical wheel (Fig. 4a) revealed that placement of apolar residues at positions 14 and 17 might enhance both Mdm2 binding and promote interactions with the membrane by lengthening the apolar helical face. Molecular modeling suggested that phenylalanine might be an ideal residue at these positions (Fig. 4b). Accordingly, we made a series of peptides with Phe residues at one or both positions (Table 2). A peptide (MP-040) with Phe at both positions (14 and 17) showed enhanced cellular activity (EC$_{50}$ = 36 nM) in the 0% FBS, 4-hour assay but was right-shifted by >20x in the 10% serum, 16- hour assay (MP-040 in Table 2), likely due to serum binding. The peptides with the extra Phe residues were either poorly soluble (12 µM or less) or poorly behaved in our solubility assay (Table 2), suggesting that they

**Table 2 | Strategic placement of charged residues in the C-terminal tail can enhance cellular activity and mitigate off-target effects**

| Peptide | C-terminus (1st position is residue 14) | Binding | Solubility | Cell EC50 (4 hr, 0% FBS) (µM) | Cell EC50 (16 hr, 10% FBS) (µM) | Proliferation EC$_{50}$ | Proliferation EC$_{50}$ | Proliferation % inhibition at 50 µM |
|---|---|---|---|---|---|---|---|---|
| | | $K_D$ (nM) | (µM) | | | HCT116 (On-target) (µM) | Ca Ski (Off-target) (µM) | Ca Ski % |
| MP-081 | -SA$_2$-NH$_2$ | 5.0 | 105 | 0.49 | 0.86 | 2.3 | 32.8 | 85.5 |
| MP-688* | -A$_6$-NH$_2$ | 21.5 | N.D. | 0.16 | 1.3 | 12.0 | 45.1 | 53.4 |
| ALRN-6924* | -A$_5$-dA-NH$_2$ | 2.46 | P.B. | 0.14 | 0.32 | 0.87 | 47.5 | 48.7 |
| MP-032 | -A$_6$-NH$_2$ | 9.45 | P.B. | 0.14 | 0.24 | 0.96 | 26.6 | 47.5 |
| MP-464 | -A$_5$-dA-NH$_2$ | 30.9 | 157 | 0.43 | 0.29 | 0.42 | >49.75 | 9.9 |
| MP-8834 | -EA$_4$-dA-NH$_2$ | 1.1 | 167 | 1.0 | 3.4 | 1.4 | >49.75 | -3.2 |
| MP-965 | -AEA$_3$-dA-NH$_2$ | 5.7 | 161 | 1.1 | 2.2 | 2.7 | >49.75 | 16.2 |
| MP-833 | -A$_2$EA$_2$-dA-NH$_2$ | 3.1 | 154 | 0.96 | 0.19 | 0.14 | >49.75 | -11.1 |
| MP-835 | -A$_3$EA-dA-NH$_2$ | <1 | 162 | 0.65 | 0.82 | 0.55 | >49.75 | -10.3 |
| MP-837 | -A$_4$E-dA-NH$_2$ | 5.7 | 160 | 0.76 | 1.67 | 0.66 | >49.75 | 8.6 |
| MP-836 | -A$_5$E-NH$_2$ | <1 | 164 | 0.55 | 1.06 | 1.05 | >49.75 | 13.2 |
| MP-897 | -(Gla)A$_4$-dA-NH$_2$ | <1 | 163 | 0.79 | 0.52 | 0.79 | >49.75 | -19.1 |
| MP-060 | -A$_5$F-NH$_2$ | 2.1 | P.B. | 0.17 | 2.7 | 0.46 | >49.75 | 18.6 |
| MP-228 | -A$_2$FA$_2$-dA-NH$_2$ | 4.2 | 2 | 0.21 | 0.69 | 2.7 | 30.7 | 68.7 |
| MP-040 | -(A$_2$F)$_2$-NH$_2$ | 27.1 | P.B. | 0.036 | 0.75 | 1.22 | >49.75 | 4.4 |
| MP-063 | -A$_2$FA$_2$-dF-NH$_2$ | 56.0 | 9 | 0.11 | 2.13 | 2.78 | >49.75 | 24.0 |
| MP-288 | -A$_2$FA$_2$-(aMe-F)-NH$_2$ | 23.7 | 12 | 0.10 | 2.68 | 2.74 | >49.75 | 5.3 |
| MP-229 | -A$_2$FA-(aMe-E)-F-NH$_2$ | 4.8 | 153 | 0.20 | 0.21 | 2.32 | 19.9 | 102.7 |
| MP-802 | -((hS)$_2$F)$_2$-NH$_2$ (early isomer) | 7.4 | P.B. | 0.23 | 0.75 | 4.58 | 19.1 | 95.2 |
| MP-803 | -((hS)$_2$F)$_2$-NH$_2$ (late isomer) | 121.3 | 1 | 0.45 | 2.49 | 2.03 | 27.0 | 84.5 |

Constant N-terminal sequence of Ac-K(N3)-βA-LTF(R8)EYWAQ(Cba)(S5)-, except for peptides marked with a asterisk which instead have an N-terminal sequence of Ac-LTF(R8)EYWAQL(S5)-. N.D.: Data not collected, P.B.: Poorly behaved; result inconclusive. Compounds were tested in duplicate.

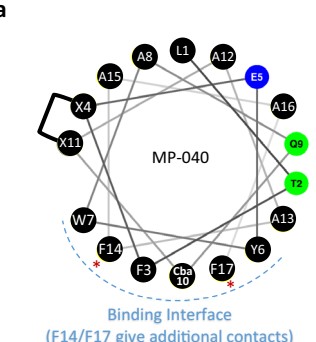

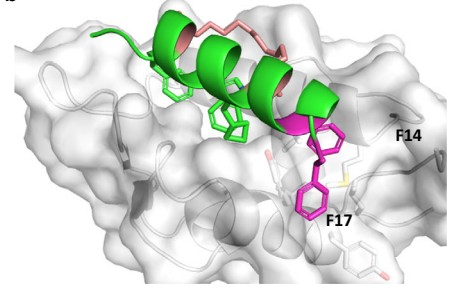

Binding Interface
(F14/F17 give additional contacts)

**Fig. 4 | Molecular modeling suggests that substitution of A14 and A17 with phenylalanine can increase binding affinity to Mdm2. a** A helical wheel representation of MP-040 sequence with introduced Phe (F14 and F17) residues highlighted with an asterisk. Black circles represent positions with apolar residues whereas those with blue circles are negatively charged positions and those with green circles are uncharged polar positions. Stapling residues are indicated with an X. **b** Structural representation of MD snapshot of MP-040 / Mdm2 complex. Mdm2 is shown as surface (grey) and bound peptide (green) is shown as cartoon with interacting residues and the two Phe residues (magenta) highlighted in sticks. The hydrocarbon linker is highlighted in salmon.

have very poor solubility. Notably, MP-220, a peptide with the di-Phe motif and an introduced single anionic residue (α-methyl Glu) had complete rescue of the shifted activity (0.20 and 0.21 µM in the 0% and 10% serum assays, respectively) and gave excellent solubility (157 µM, Table 2). However, it should be noted that this peptide had a narrow on-target index with an off-target EC$_{50}$ value in the Ca Ski proliferation assay of 19.9 µM, suggesting that further optimization is required for this sequence. In contrast to the improvements in peptide behavior

imparted by the introduction of a single anionic residue, the addition of multiple copies of a polar—but non-ionic—residue (homoSer) failed to rescue solubility and serum shift (the -((hS)$_2$F)$_2$-NH$_2$ peptides, Table 2). This highlights the charge requirement for enhancing solution behavior. Overall, these results suggest that it is possible to introduce additional apolar residues into the tail to improve interactions with Mdm2 and/or the cell membrane but that these need to be balanced with negative charge to maintain drug-like properties.

**Optimizing staple type and number significantly mitigates off-target activity.** To explore the effect of staple number and type on peptide properties, we made a series of peptides with alternative crosslinks. Within the context of MP-081 (-SAA tail), replacing the olefin staple with a triazole crosslink (Fig. 5a, MP-707) resulted in high affinity binding (1.1 nM) but gave poorer cellular potency, especially in the presence of 10% serum (13.6 µM, Fig. 5a). On the other hand, adding a second staple at the N-terminus through a K1 to E5 lactam bridge (Fig. 5a, MP-002) gave moderate cellular potency (2.2 µM, 10% serum, Fig. 5a) but also gave poor behavior in our solubility assay, presumably due to the removal of the negative charge. We also examined an MP-464 (polyAla) analog containing a patent-derived di-alkyne staple[24]. This cross-link should be very rigid due to the conjugated triple bonds (see molecular structure in Fig. 5a). The resulting peptide (MP-616), containing both the rigid staple and the $A_5$-dA-NH$_2$ tail, had very good solubility (128 µM) and good cellular activities: 480 nM in the 10% serum, 16-hour p53 assay and 124 nM in the HCT116 cell proliferation assay (Fig. 5a). Importantly, both the lactam peptide and the di-alkyne stapled peptide proved to be remarkably devoid of off-target toxicity across our cell proliferation panel (Fig. 5b), which differentiated them from other stapled peptides examined (Figs. 2c and S3), as well as advanced Mdm2 small molecule antagonists, MK-4688[28] and AMG 232[29] (Fig. 5b). The way the double-stapled and di-alkyne stapled peptides avoid off-target toxicities is not clear. However, this result may be related to reduced promiscuous binding by restricting conformational freedoms – either of the staples themselves or of the peptide backbone.

**Peptides with polyAla tails have improved in vivo activity versus ATSP-7041**
To understand whether improvements in in vitro peptide activity translated to the in vivo setting, we compared ATSP-7041 head-to-head with two polyAla containing peptides in a mouse SJSA-1 xenograft model. As one candidate, we selected ALRN-6924 since it was of interest to explore the in vivo performance of the clinical molecule. For the other molecule, we chose the di-alkyne stapled peptide, MP-616 since it had a different macrocycle and a very clean off-target profile. ATSP-7041 had clear tumor growth inhibition (~ 33% tumor growth inhibition) when dosed intravenously every third day at 30 mg/kg but was outperformed by both ALRN-6924 and MP-616, which had large and significant improvements (~ 66% tumor growth inhibition, Fig. 6). Although not examined here, it would be interesting to understand if the clean off-target profile for MP-616 translates into fewer side-effects in vivo, especially considering the poly-pharmacology observed with ALRN-6924 at higher doses in the counterscreen line, Ca Ski (Table 2).

**Combining favorable attributes resulted in a peptide with low-nanomolar potency.** In addition to improvements from appending the polyalanine tails, judicious placement of negative charge, and the di-alkyne staple, we also noted that an L1 to azido-lysine substitution resulted in improved cellular potency in an ALRN-6924 analog (MP-062, 117 nM in 10% serum, 16 hours, Supplementary Data 1). Accordingly, we combined this improvement with the di-alkyne staple and an optimized C-terminal tail to arrive at MP-444: Ac-K(N3)-TF(X)EYWAQ(Cba)(X)EAFAAF-NH$_2$ (where X = the di-alkyne stapling positions), which had good solubility (167 µM), a binding $K_D$ of 1.5 nM, cellular EC$_{50}$s of 51 and 153 nM (4 hours, 0% serum and 16 hours, 10% serum, respectively) and a cell proliferation EC$_{50}$ of 60 nM against HCT116 (Supplementary Data 1). This peptide also had some slight off-target effects, although the on-target index was ~1000x compared to the Ca Ski proliferation (EC$_{50}$ > 50 µM but with 39.5% growth inhibition at 50 µM). Dramatic improvements were also seen with a di-alkyne stapled peptide with an a-methyl glutamic acid in the polyAla tail. Specifically, MP-467 (Ac-K(N3)-(betaAla)-LTF(X)EYWAQ(Cba)(X)AA(a-methly-Glu)AA(DAla)-NH2, Supplementary Data 1) showed a potency

of 30 nM in the serum-containing cell reporter assay and an EC$_{50}$ = 13 nM in the HCT116 cell proliferation assay (Supplementary Data 1). Remarkably, this peptide was completely devoid of proliferation effects in the Ca Ski control line (Fig. 7, EC$_{50}$ ≫ 50,000 nM; no measurable inhibition at 50,000 nM). Overall, this peptide was the most potent molecule observed to date in our HCT116 cell proliferation assay and with an on-target index of >3800x. Furthermore, in terms of cell proliferation potency, this peptide represents a 292x improvement over our starting point (azide-ATSP-7041) and an 80x improvement over ALRN-6924 (Fig. 7). Finally, MP-467 also greatly outperformed the advanced Mdm2 small molecule antagonists MK-4688 and AMG 232, both in terms of their on-target profiles (Fig. 4b vs Fig. 6) and 16-hour 10% serum p53 activities (>10x superior potency, Supplementary Data 1).

**Application of peptide design insights to a stapled D-peptide improves its cellular potency by 100-fold and removes off-target activity**
Next, we sought to understand whether the lessons learned from the ATSP-7041 library could be used to improve peptides from an independent series. We focused on the $^d$PMI-d(6-10 staple) molecule (MP-769), a previously described[30] all-D stapled peptide with Mdm2 antagonistic activity. This molecule was an ideal test case since it had innate hyper-stability (due to its all-D nature) and verified binding but exhibited poor cellular activity (e.g., inactive in 10% serum at 16 hr, Fig. 8) and suffered from LDH release and counterscreen liabilities, likely due to its multi-cationic nature (free N-terminus, K9 and R12, Fig. 8). It also had an amphipathic character (Fig S5) which we judged would be beneficial for maintaining peptide solubility. We first sought to remove the positive charges (vide supra). Using ATSP-7041 as an amphipathic template with good cell permeability and activity, we noted the presence of an acetylated N-terminus and Ser and Gln residues on its polar helical face (Fig S5). To mimic this topology, we acetylated the N-terminus of MP-769 and introduced the K9Q and R12S substitutions, changes that produced an improved peptide (MP-797) that was inactive in the LDH-release and counterscreen assays (EC$_{50}$s > 50 µM, Fig. 8b). We noted that this peptide had good solubility (167 µM, Fig. 8b), a value that was dependent on the molecule having a net negative (-1) charge since a peptide with the same sequence, but with a net-neutral charge due to its free N-terminus, had poor solubility (1 µM, MP-795, Supplementary Data 1). The cellular activity of MP-797 was also improved by 3-fold (EC$_{50}$ = 3.6 µM, Fig. 8b) in the 0% serum, 4-hour assay. However, the compound still had weak cellular efficacy (EC$_{50}$ > 50 µM) under 10% serum conditions and displayed a poor on-target cell proliferation profile (Fig. 8b). We then appended poly-d-alanine tails of different lengths. Consistent with the ATSP-7041 series, the (D-Ala)$_6$ C-terminal tail provided the best profile for the all-D series (Supplementary Data 1). Specifically, MP-042 had excellent solubility (150 µM) and a cellular activity of 0.28 µM in the 10% serum, 16-hour p53 assay, representing an overall improvement of >178-fold compared to the initial parent peptide (Fig. 8b). However, despite being devoid of LDH and counterscreen activity, this peptide showed a narrow therapeutic index (~ 20-fold) with proliferation potencies of 1.5 and 31 µM in HCT116 and Ca Ski, respectively (Fig. 8b). This may result from the significant conformational freedom available to an i to i + 4 stapled peptide, which may allow for promiscuous binding to a variety of targets. Finally, decreasing (3x D-Ala, MP-230) or further increasing (9x D-Ala, MP-1230) the length of the tail led to compounds with poorer cellular efficacy, and for the 9x D-Ala tail peptide, poorer solubility (Supplementary Data 1), similar to observations in the ATSP-7041 series (Table 2).

To address the off-target effects observed with the i to i + 4 stapled all-D peptide (MP-042), we explored whether switching a longer (i to i + 7) and more rigid (di-alkyne) linkage could mitigate these effects. Accordingly, we elected to apply our emerging design rules to

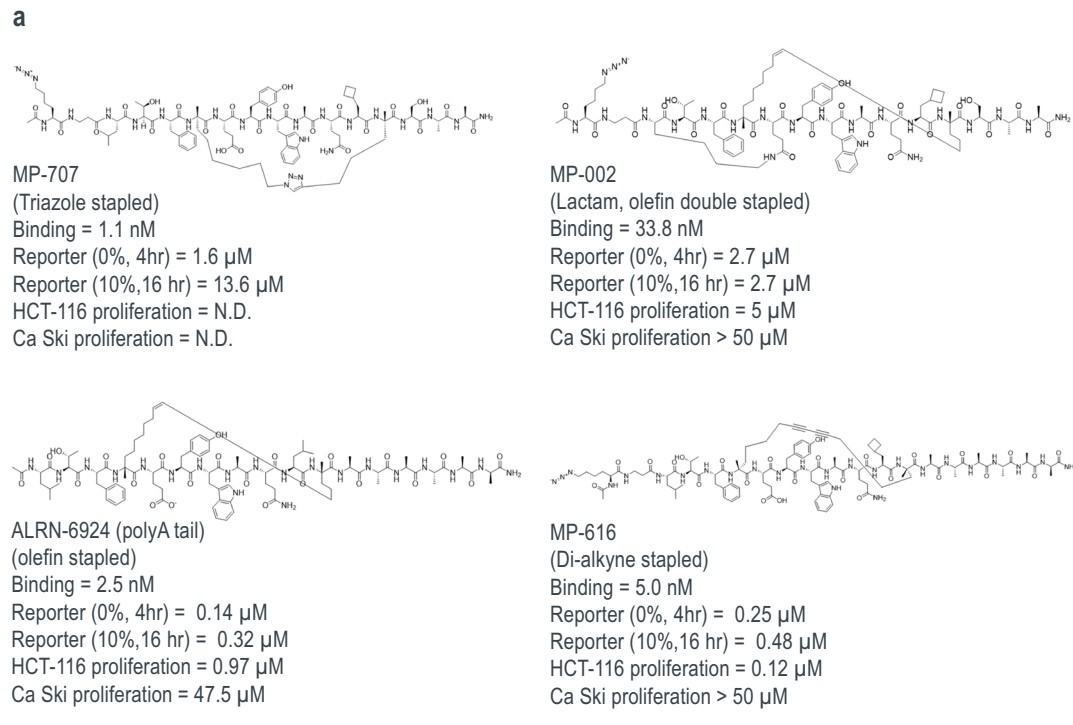

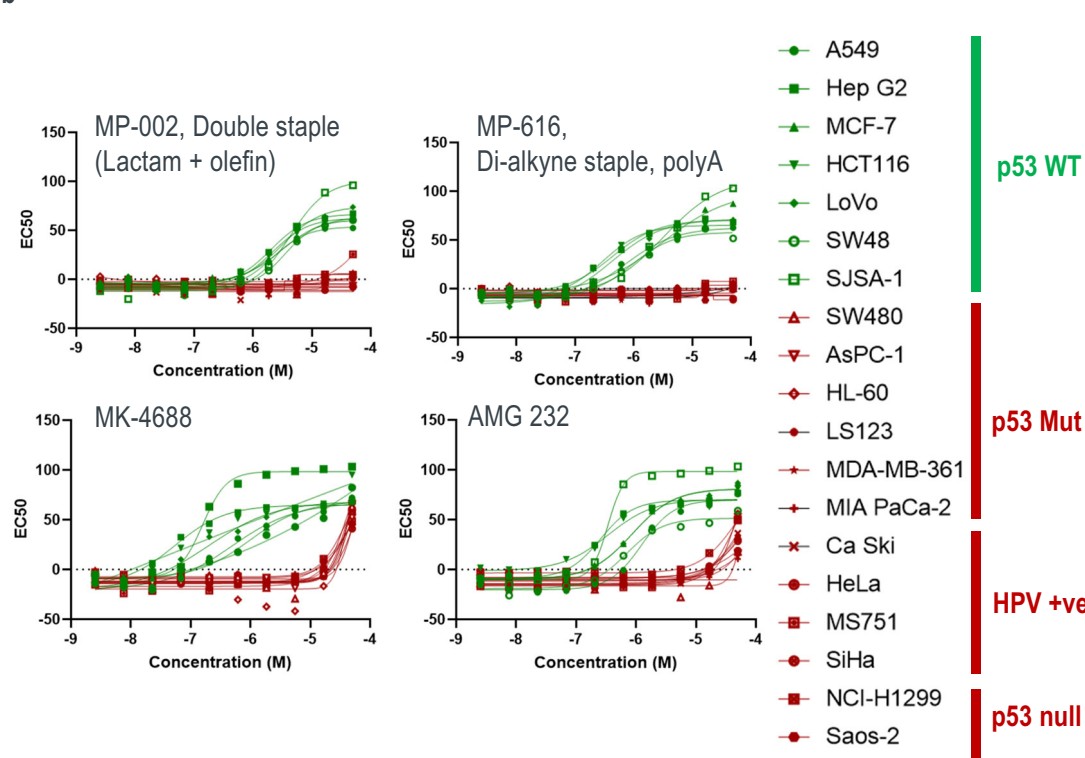

**Fig. 5 | Optimizing staple type and number significantly mitigates off-target activity. a** Structure and profile of peptides with alternative cross-linking strategies with ARLN-6924 shown as a comparator. **b** Proliferation profiling against a panel of cancer cell lines that are either p53 WT (green) and p53 defective (i.e., p53 null, mutant, or p53-depleted through HPV infection, red) reveals that peptides with alternative crosslinking strategies (double stapled or di-alkyne stapled) produces molecules that were devoid of off-target proliferation effects whereas advanced small molecules (MK-4688 and AMG 232) showed significant off-target toxicities. Compounds were tested in duplicate.

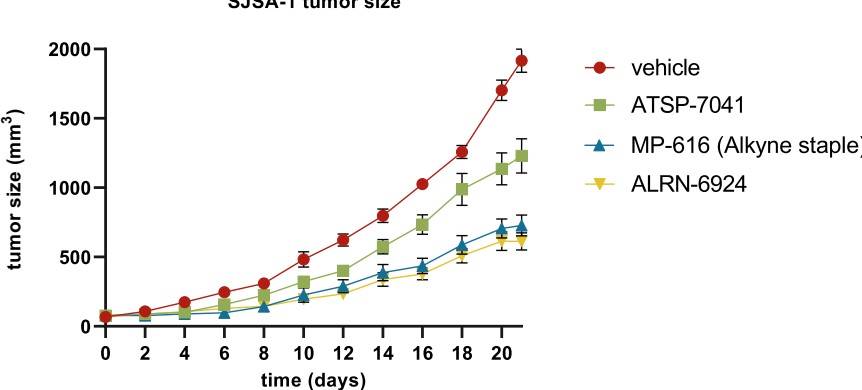

**Fig. 6 | Peptides with polyAla tails have improved in vivo activity versus ATSP-7041 in a SJSA-1 xenograft model.** Peptides were injected intravenously twice a week with vehicle control or peptide at 30 mg/kg for twenty-five days (ten mice per group). Mean values are plotted with error bars representing standard error of the mean.

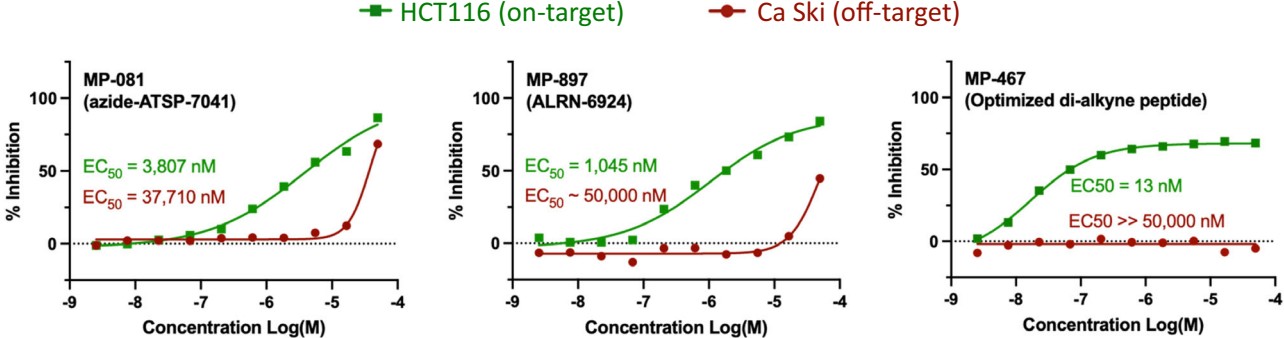

**Fig. 7 | Combining favorable attributes resulted in a peptide with low-nanomolar potency.** Appending MP-081 with a C-terminal polyAla tail, judicious placement of an additional anionic residue and replacement of the conventional olefin staple with a di-alkyne staple resulted in an optimized peptide (MP-467) whose cell proliferation profile greatly outperformed both MP-081 and ALRN-6924, both in terms of on-target (HCT116) potency and off-target (Ca Ski) toxicity. Compounds were tested in duplicate.

[d]PMI-d(5-12 staple) (MP-770), a previously reported stapled all-D peptide with an i to i + 7 olefin staple[30], with modest cellular potency that overlapped with its counterscreen activity (Fig. 8b). To address these deficiencies, we i) introduced a K9Q substitution along with N-terminal acetylation to remove the positive charges; ii) introduce the (D-Ala)$_6$ polyalanine C-terminal tail to optimize solubility and helicity; iii) replaced the i, i + 7 olefin macrocycle with the di-alkyne staple. These changes resulted in MP-793 which had ultra-tight binding ($K_D$ < 1 nM, Fig. 8b) and a > 102x higher potency (0.3 µM, Fig. 8b) in the 10% serum, 16-hour assay (versus MP-770, the i to i + 7 olefin parent). Notably, compared to the optimized 6-10 stapled peptide (MP-042), MP-793 had a > 2x improved cell proliferation potency and a greatly improved on-target index with proliferation EC$_{50}$s of 0.65 and >50 µM against HCT116 and Ca Ski, respectively (HCT116 % inhibition at 50 µM = 0.5% Fig. 8b). This result is consistent with the excellent on-target profiles achieved with the di-alkyne stapled L-peptides (e.g., MP-467 and MP-616).

## Discussion

Although stapled peptides hold therapeutic promise to tackle highly validated but historically intractable targets, a corresponding approved therapeutic has yet to be realized. This is not surprising since it is a relatively new class of molecules, and its discovery and developmental challenges are still being defined. The work described herein is the most exhaustive report to-date to explore structure-activity relationships of a large library (> 400) of staple peptides. These studies have resulted in an improved understanding for how to generate a lead stapled peptide against an intracellular target. Learnings from the ATSP-7041 library resulted in molecules that had improved binding, cellular activity (up to 292x improvement), and, in some cases, undetectable off-target toxicity. Critically, a select lead compound also gave improved in vivo activity compared to ATSP-7041, a molecule that had already been extensively optimized[13]. Using the same target protein (Mdm2), these emerging 'design rules' were then validated in a second chemically distinct series, resulting in removal of off-target toxicities and an improvement in cellular activity of >100x.

To summarize the knowledge gained, we outline our general workflow for designing a lead stapled peptide against an intracellular target. In practice, the order of steps proposed is unlikely to be entirely linear as drug discovery often follows a meandering path where the improvements to one property often results in the worsening of another. As such, certain steps may need to be revisited before arriving at a lead molecule. Nevertheless, the stepwise protocol listed below should serve as a general framework:

1. **Identification of high-affinity binders:** Lead identification initiates with the discovery of a high-affinity a-helical binder against a target of interest. Phage and mRNA display are mature technologies that allow for the selection of high affinity sequences with high probability[31–33]. Some systems[32,34,35] enable the use of staples within the primary screen, which can aid in the discovery of helical binders. Starting points can also be discovered through rational-design by leveraging a-helices that naturally mediate PPIs[36,37]. Post-screening, affinity maturation may be achieved through substitution of interfacial residues with both natural and

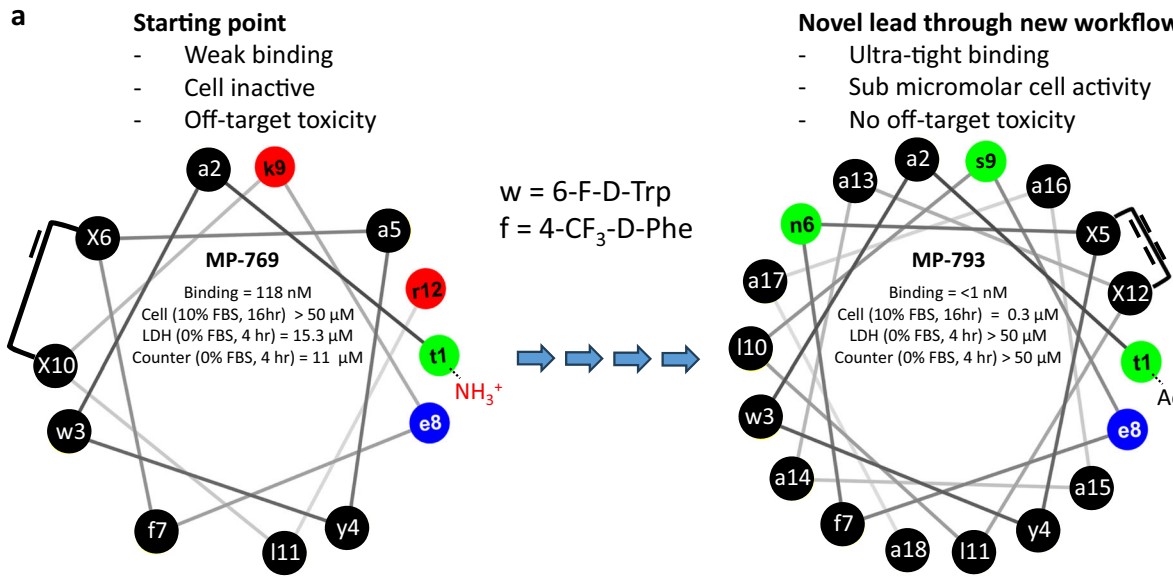

**b**

Detailed progression from MP-769 to MP-793, ↓ = modification sites

<u>i to i + 4 all-D stapled peptides</u>

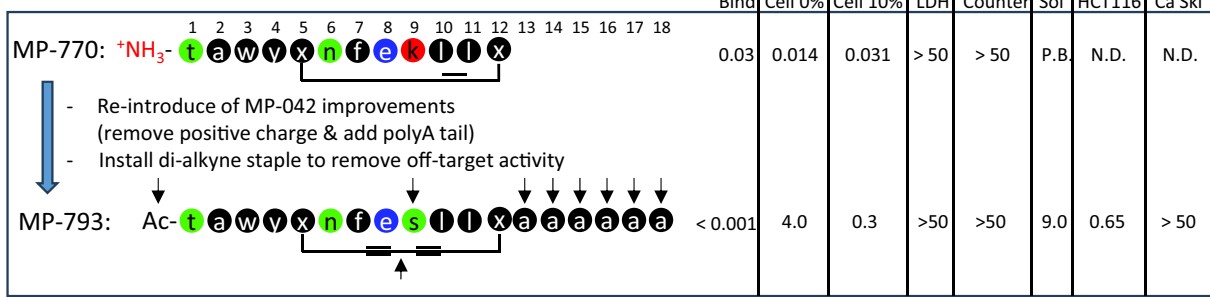

| | Bind | Cell 0% | Cell 10% | LDH | Counter | Sol | HCT116 | Ca Ski |
|---|---|---|---|---|---|---|---|---|
| MP-769 | 0.12 | 10 | > 50 | 15 | 11 | 74 | N.D. | N.D. |
| MP-797 | 0.26 | 3.6 | >50 | >50 | >50 | 167 | 9.1 | 30.4 |
| MP-042 | 0.03 | 0.6 | 0.3 | >50 | >50 | 150 | 1.5 | 31 |

Decrease conformational freedom by switching to a literature i to i + 7 stapled peptide

<u>i to i + 7 all-D stapled peptides</u>

| | Bind | Cell 0% | Cell 10% | LDH | Counter | Sol | HCT116 | Ca Ski |
|---|---|---|---|---|---|---|---|---|
| MP-770 | 0.03 | 0.014 | 0.031 | > 50 | > 50 | P.B. | N.D. | N.D. |
| MP-793 | < 0.001 | 4.0 | 0.3 | >50 | >50 | 9.0 | 0.65 | > 50 |

- Re-introduce of MP-042 improvements (remove positive charge & add polyA tail)
- Install di-alkyne staple to remove off-target activity

non-natural amino acids. At this stage, we recommend removing all cationic residues that are non-critical to binding. This is important as peptides that are rich in both apolar residues and have a net positive charge often promote cell toxicity and false-positive readouts in cellular assays, as shown by previous work[17] and this study (e.g. the net positively charged peptides in Supplementary Table 1).

2. **Selection of staple position:** With a high affinity helical binder in-hand, the next task is to identify the optimal stapling position. To be exhaustive, a staple-scan is preferred where all possible positions are surveyed. Although many staple types exist, our suggestion is to limit the initial analogs to a single chemistry: olefin staples resulting from ring closing metathesis of (R)-2-(7′-octenyl)alanine and (S)-2-(4′-pentenyl)alanine at the i and i + 7

**Fig. 8 | Application of peptide design insights to a stapled D-peptide improves its cellular potency by 100-fold and removes off-target activity. a** helical wheel representations illustrating how the application of design rules derived from the ATSP-7041 series to an distinct all-D series transformed a cellularly inactive peptide with off-target toxicity ($^d$PMI-δ(6-10), also known as MP-769) to a peptide that is devoid of off-target toxicity and with > 100x improvement in cellular activity. **b** Linear sequence representation of the step-wise improvements made to MP-769 resulting in MP-793. Black arrows highlight the peptide changes. Residues colored in black are apolar, those in red are cationic, those in blue are

anionic, and those in green are uncharged polar. Values in the right hand table are in micromolar. Bind = binding affinity, Cell 0% = $EC_{50}$ value in the cellular assay (4 hour incubation, 0% serum), Cell 10% = $EC_{50}$ value in the cellular assay (16 hour incubation, 10% serum), LDH = $EC_{50}$ value in the LDH assay (4 hour incubation, 0% serum), Counter = $EC_{50}$ value in the cellular counterscreen assay (4 hour incubation, 0% serum), Sol = solubility, HCT116 = $EC_{50}$ value in the on-target cell proliferation assay, Ca Ski = $EC_{50}$ value in the off-target cell proliferation assay. Compounds were tested in duplicate.

positions, respectively. This initial chemistry is suggested as i, i -> 7 staples give better helical stabilization and superior proteolytic resistance versus i, i -> 4 staples. In addition, cost-effective commercial building blocks are readily available and the reaction results in high yields. With an initial library of stapled peptides in-hand, routine screening of binding, solubility, cellular activity, and stability should begin (although this latter property is not of concern if employing an all-D peptide). These criteria should be considered in aggregate for selecting the final stapling position. The position of the staple will also influence the peptide's initial and potential amphipathicity[23], a property that may be important for peptide solubility and aqueous behavior. For example, the binding interface of the ATSP-7041 peptide consists of three spatially contiguous apolar residues (F, W, and Cba) and is flanked by the apolar staple. This presentation makes it straightforward to engineer an amphipathic helix by placing hydrophilic residues on the solvent facing side, an arrangement that balances the dual needs for membrane interaction (for permeability) with good aqueous behavior.

3. **Optimization of peptide length:** Next, we recommend optimizing peptide length. Although this work provides examples where extending peptide length gives improved cellular activities, examinations of shorter peptides may also be fruitful. To lengthen a peptide, we recommend starting with a series of N- and C-terminal extensions by appending varying numbers of alanines, a slightly apolar residue with high helical propensity.

4. **Optimization of staple type and number:** Optimization of staple type and number may give improved peptide performance in a few ways. First, target affinity may be improved by stabilizing the peptide in its binding conformation. Peptide stability may also be improved by preventing proteases from engaging with exposed peptide bonds. In addition, peptides that are better locked into the a-helical conformation will make fewer hydrogen bonds with water as their backbone amide and carbonyl moieties will be engaged with each other, as part of the a-helical architecture. As a result, improved membrane permeabilities may result from a smaller desolvation penalty associated with traversing the membrane bilayer. As shown by this work, optimization of peptide type (e.g., the di-alkyne staple in MP-616) or number (e.g., the double-stapled MP-002) may also decrease non-specific toxicities, perhaps due to restricted conformational sampling of the peptide leading to decreased interactions with off-target biomolecules. A simple approach for restricting conformational heterogeneity is through the introduction of a rigid staple such as the dialkyne linkage reported by Aileron[24] and used here. Another strategy is to employ multiple staples. Options include a variety of chemistries including but not limited to hydrocarbon staples via RCM, lactam bridges (e.g. MP-002), proline-locked peptides[38], hydrogen bond surrogates[39], and a variety of other staple types[40] including, crosslinked cysteine residue, and triazole linkers. Possibilities also encompass a variety of residue spanning topologies (e.g., i -> i + 3, i -> i + 4, i -> i + 7, etc.)[5]. In addition peptides with multiple staples may have double[41], stitched[30,42] and cross-stitched arrangements[43]. It may be of interest to explore whether macrocyclic strategies that lock the entire peptide length

into the helical conformation through a network of crosslinks from termini to termini would be optimal. Finally, although not macrocyclic in nature, conformational restriction can also be achieved through the introduction of residues with high helical propensity, including a-methylated residues. This latter strategy may also confer additional protease resistance.

5. **Final sequence optimization:** once peptide length and stapling strategies are finalized, peptide properties may be further optimized through sequence exploration. First, if the helix has been lengthened, there may be opportunities to improve binding affinity by introduction of sidechains within the extended region that productively engage the target surface (e.g., the introduction of the Phe residues in MP-040 and analogs such as MP-444). As observed here, once the target binding interface is optimized, stapled peptides can have extensive flexibility for sidechain identity at the solvent-facing positions. Sidechain exploration in this region is critical for optimizing peptide permeability while maintaining peptide solubility. We suggest aiming for a helix with both apolar and polar helical faces. In such amphipathic peptides, the apolar side is required to promote peptide permeability through productive membrane interactions. However, as seen from our library analysis, sequences should not be made too hydrophobic as they can result in cell toxicity, poor solubilities, and shifts in cellular activity in the presence of serum. Overtly hydrophobic sequences can potentially be rescued by judicious placement of hydrophilic residues. Specifically, placement of an acidic residue is preferred as basic residues have liabilities (*vide supra*) and non-charged side chains had minimal effects on peptide solubility, although these latter residues may be required as part of the hydrophilic face. Examples reported here show how a single Glu residue can enhance solubility (e.g., MP-444) and reduce off-target toxicities (e.g., the peptides with multiple anionic charges in Table 2). An extensive library of analogs, including non-natural amino acids, may be required to arrive at a peptide with the optimal balance of the nature, number, and arrangement of apolar and polar residues.

Although the current work has provided valuable insights, we would like to underscore some limitations. First, although the application of lessons gained from the ATSP-7041 library resulted in significant improvements in a distinct chemical series (the $^d$PMI-d peptides), it is important to highlight that both series are directed towards the same target(s) (Mdm2/X). Thus, future studies should aim to understand if the design rules uncovered here can make similar impacts to stapled peptides against distinct targets. Indeed, the workflow we propose here is likely to be most applicable to targets that - like Mdm2/X - have a hydrophobic PPI interface that is addressable with a helical motif. Indeed, we expect that not all targets will have geometries that will accommodate or otherwise be suitable for high-affinity binding to a helical segment. For targets where the PPI interface is hydrophilic, it will be interesting to explore whether our generalized workflow can be altered in a corresponding manner. In particular, one might focus on optimizing the polar surface for binding affinity while installing hydrophobic residues on the opposing side of the helix to gain cell permeability. In addition, it will

be interesting to see if judicious placement of anionic residues and optimization of staple type/number is a generalizable approach for removing stapled peptide toxicities. Another limitation of our study is that although some of the peptides reported here appear to be devoid of off-target effects, those observations come from a narrow set of experiments. Conducting proteome-wide surveys (using MS-CETSA or related techniques) would be of interest to understand the extent to which these molecules can engage with off-targets. In addition, although we have increased our understanding for identifying lead peptides, drug hunters will appreciate that converting these leads to clinical candidates may require further exploration of the chemical space. In particular, in-depth studies of stapled peptide formulation and pharmacokinetic properties were not explored here and have yet to be reported elsewhere. Optimizing these properties may require extensive efforts. Furthermore, understanding the ways in which these molecules are similar to—and distinct from—small molecules will help inform on dosing regimens and routes of administration. Achieving oral bioavailability through sequence and/or formulation optimization would expand the disease-states stapled peptides can be applied to.

Finally, the specific molecules identified here may represent interesting therapeutic candidates. For example, a di-alkyne stapled peptide appeared to give equivalent in vivo performance to ARLN-6924, a molecule that is currently undergoing clinical trials. Indeed, the optimized di-alkyne staple peptide (MP-467) may be a preferred clinical molecule given its greatly improved cellular potency (80x vs ALRN-6924) and superior on-target profile, which may translate to fewer adverse effects in humans. The all-D lead peptide (MP-793) would be interesting to explore as well given its inherent hyperstability. However, stoichiometric inhibitors engaging with the Mdm2 protein have only enjoyed limited clinical success so far, perhaps due to the built-in feedback mechanism that upregulates Mdm2 protein levels to limit therapeutic efficacy, including that of ARLN-6924. Accordingly, successfully addressing Mdm2 may require a targeted protein degradation approach. In fact, application of this paradigm to Mdm2 small molecules appears to bring improved efficacy[44,45] and an Mdm2 degrader has advanced to the clinic[46]. Dual degrading of Mdm2/X may be therapeutically differentiated, especially in cancers that are MdmX upregulated. Thus, converting optimized Mdm2/X stapled peptide antagonists into degraders is worth pursuing, especially since dual degradation will likely be challenging with a small molecule ligand. Design ideas for such peptide-based degraders could include hybrid heterobifunctional molecules consisting of a stapled peptide fused to an E3 ligase small molecule ligand. On the other hand, insights from this manuscript suggest that Mdm2/X-directed stapled peptides could also be converted into Mdm2-degrading molecular glues. Specifically, given the tolerability for substitutions outside of the binding interface, researchers could explore substitutions on the solvent exposed side of the peptide to recruit an E3 ligase to degrade Mdm2/X. Of course, the same paradigm could also be applied to use Mdm2 as the recruited E3 ligase towards neosubstrates of therapeutic value.

## Methods

Biological assays were run in at least duplicates (biological replicates). Duplicate values were typically within 1.5x of each other. If the duplicates were >3x different, a third replicate was run. At this point, if all collected values were within 3x of the geometric mean, no further measurements were typically taken. Otherwise, further measurements were taken until this requirement was met. Values were reported as geometric means.

### Mdm2 protein production

For use in the peptide binding assay, a human Mdm2 1–125 sequence was cloned into a pNIC-GST vector. The TEV (tobacco etch virus) cleavage site was changed from ENLYFQS to ENLYFQG to give a fusion protein with the following sequence:

MSDKIIHSPILGYWKIKGLVQPTRLLLEYLEEKYEEHLYERDEGDKWR NKKFELGLEFPNLPYYIDGDVKLTQSMAIIRYIADKHNMLGGCPKERAEIS MLEGAVLDIRYGVSRIAYSKDFETLKVDFLSKLPEMLKMFEDRLCHKTYLN GDHVTHPDFMLYDALDVVLYMDPMCLDAFPKLVCFKKRIEAIPQIDKYL KSSKYIAWPLQGWQATFGGGDHPPKLEVLFQGHMHHHHHHSSGVDLG-TENLYFQGMCNTNMSVPTDGAVTTSQIPASEQETLVRPKPLLLKLLKSVG AQKDTYTMKEVLFYLGQYIMTKRLYDEKQQHIVVYCSNDLLGDLFGVPSF SVKEHRKIYTMIYRNLVVVNQQESSDSGTSVSEN-.

The corresponding plasmid was transformed into BL21 (DE3) Rosetta T1R Escherichia coli cells and grown under kanamycin selection. Bottles of 750 mL Terrific Broth, supplemented with appropriate antibiotics and 100 µL of antifoam 204 (Sigma-Aldrich, St. Louis, MO, USA, were inoculated with 20 mL seed cultures grown overnight. The cultures were incubated at 37 °C in the LEX system (Harbinger Biotech, Toronto, Canada) with aeration and agitation through the bubbling of filtered air through the cultures. The LEX system temperature was reduced to 18 °C when culture OD600 reached 2, and the cultures were induced after 60 min with 0.5 mM IPTG. Protein expression was allowed to continue overnight. Cells were harvested by centrifugation at 4000× g, at 15 °C for 10 min. The supernatants were discarded and the cell pellets were resuspended in a lysis buffer (1.5 mL per gram of cell pellet). The cell suspensions were stored at −80 °C before purification work.

The re-suspended cell pellet suspensions were thawed and sonicated (Sonics Vibra-Cell, Newtown, CO, USA) at 70% amplitude, 3 s on/off for 3 min, on ice. The lysate was clarified by centrifugation at 47,000× g, 4 °C for 25 min. The supernatants were filtered through 1.2 µm syringe filters and loaded onto the AKTA Xpress system (GE Healthcare, Fairfield, CO, USA). The purification regime is briefly described as follows. The lysates were loaded onto a 1 mL Ni-NTA Superflow column (Qiagen, Valencia, CA, USA) that had been equilibrated with 10 column volumes of wash 1 buffer. Overall buffer conditions were as follows: Immobilized metal affinity chromatography (IMAC) wash 1 buffer—20 mM HEPES ((4-(2-hydroxyethyl)-1-piperazineethanesulfonic acid), 500 mM NaCl, 10 mM Imidazole, 10% (v/v) glycerol, 0.5 mM TCEP (Tris(2-carboxyethyl)phosphine), pH 7.5; IMAC wash 2 buffer—20 mM HEPES, 500 mM NaCl, 25 mM Imidazole, 10% (v/v) glycerol, 0.5 mM TCEP, pH 7.5; IMAC Elution buffer—20 mM HEPES, 500 mM NaCl, 500 mM Imidazole, 10% (v/v) glycerol, 0.5 mM TCEP, pH 7.5. The sample was loaded until air was detected by the air sensor, 0.8 mL/min. The column was then washed with wash 1 buffer for 20 column volumes, followed by 20 column volumes of wash 2 buffer. The protein was eluted with five column volumes of elution buffer. The eluted proteins were collected and stored in sample loops on the system and then injected into gel filtration (GF) columns. Elution peaks were collected in 2 mL fractions and analyzed on SDS-PAGE gels. The entire purification was performed at 4 °C. Relevant peaks were pooled, TCEP was added to a total concentration of 2 mM. The protein sample was concentrated in Vivaspin 20 filter concentrators (VivaScience, Littleton, MA, USA) at 15 °C to approximately 15 mg/mL. (<18 kDa−5 K MWCO, 19–49 kDa−10 K MWCO, >50 kDa−30 K MWCO). The final protein concentration was assessed by measuring absorbance at 280 nm on Nanodrop ND-1000 (Thermo Fisher, Waltham, MA, USA). The final protein purity was assessed on SDS-PAGE gel. The final protein batch was then aliquoted into smaller fractions, frozen in liquid nitrogen and stored at −80 °C.

### Mdm2 binding by competitive fluorescence anisotropy assay

Purified Mdm2 (1-125) protein was titrated against 50 nM carboxyfluorescein (FAM)-labeled 12/1 peptide (FAM-RFMDYWEGL-NH2). The dissociation constants for titrations of Mdm2 against FAM-labeled 12/1 peptide were determined by fitting the experimental data to a 1:1

binding model equation shown below:

$$r = r_o + (r_b - r_o) \times \frac{(K_d + [L]_t + [P]_t) - \sqrt{(K_d + [L]_t + [P]_t)^2 - 4[L]_t[P]_t}}{2[L]_t}$$

(1)

[P] is the protein concentration (Mdm2), [L] is the labeled peptide concentration, r is the anisotropy measured, $r_o$ is the anisotropy of the free peptide, $r_b$ is the anisotropy of the Mdm2–FAM-labeled peptide complex, $K_d$ is the dissociation constant, $[L]_t$ is the total FAM labeled peptide concentration, and $[P]_t$ is the total Mdm2 concentration. The apparent $K_d$ value for FAM-labeled 12/1 peptide against Mdm2 was determined to be 13.0 nM respectively. This value was then used to determine apparent $K_d$ values of the respective competing ligands in subsequent competition assays in fluorescence anisotropy experiments.

Mdm2 competition experiments were performed with their respective concentrations held constant at 250 nM in the presence of 50 nM of FAM-labeled 12/1. The competing molecules were then titrated against the complex of the FAM-labeled peptide and protein. Apparent Kd values were determined by fitting the experimental data to equations shown below:

$$r = r_o + (r_b - r_o) \times \frac{2\sqrt{(d^2 - 3e)}\cos(\theta/3) - 9}{3K_{d1} + 2\sqrt{(d^2 - 3e)}\cos(\theta/3) - d}$$

(2)

$$d = K_{d1} + K_{d2} + [L]_{st} + [L]_t - [P]_t$$

(3)

$$e = ([L]_t - [P]_t)K_{d1} + ([L]_{st} + [P]_t)K_{d2} + K_{d1}K_{d2}$$

(4)

$$f = -K_{d1}K_{d2}[P]_t$$

(5)

$$\theta = \text{ar} \cos\left[\frac{-2d^3 + 9de - 27f}{2\sqrt{(d^2 - 3e)^3}}\right]$$

(6)

$[L]_{st}$ and $[L]_t$ denote labeled ligand and total unlabeled ligand input concentrations, respectively. $K_{d2}$ is the dissociation constant of the interaction between the unlabeled ligand and the protein. In all competition experiments, it is assumed that $[P]t > [L]_{st}$, otherwise considerable amounts of free labeled ligand would always be present and would interfere with measurements. $K_{d1}$ is the apparent $K_d$ for the labeled peptide used and has been experimentally determined as described in the previous paragraph. The FAM-labeled peptide was dissolved in dimethyl sulfoxide (DMSO) at 1 mM and diluted into experimental buffer. Readings were carried out with an Envision Multilabel Reader (PerkinElmer). Experiments were carried out in PBS (2.7 mM KCl, 137 mM NaCl, 10 mM Na2HPO4 and 2 mM KH2PO4 (pH 7.4)) and 0.001% Tween-20 buffer. Curve-fitting was carried out using Prism 4.0 (GraphPad). To validate the fitting of a 1:1 binding model we carefully ensured that the anisotropy value at the beginning of the direct titrations between Mdm2 and the FAM-labeled peptide did not differ significantly from the anisotropy value observed for the free fluorescently labeled peptide. Negative control titrations of the ligands under investigation were also carried out with the fluorescently labeled peptide (in the absence of Mdm2) to ensure no interactions were occurring between the ligands and the FAM-labeled peptide. In addition, we ensured that the final baseline in the competitive titrations did not fall below the anisotropy value for the free FAM-labeled peptide, which would otherwise indicate an unintended interaction between the ligand and the FAM-labeled peptide to be displaced from the

Mdm2 binding site. Measurements were taken a minimum of three times (biological replicates) and values are reported as geometric means of the replicates.

## Solubility assay
Kinetic solubility was determined by diluting 10 mM stock solution of the peptides in DMSO in PBS buffer at pH 7 to make test solutions. The standard reference solutions were prepared by diluting the 10 mM stock in 10:80:10 solution of acetonitrile:methanol:DMSO to 100 µM. Standard solution and test solutions were analyzed using Agilent 1290 UPLC/DAD system after filtering the test solutions through a 0.45 µm polypropylene filter.

The solubility value is calculated by the following equation:
Solubility [µM] = (Peak area of sample/ Peak area of standard) (Standard concentration)

## Whole cell homogenate stability
Peptides at a concentration of 1 µM were incubated at 37 °C with HCT116 whole cell homogenates prepared from 1 million lyzed cells/mL. The reaction was stopped at 0, 1, 2, and 4 hours and 22 hours with an organic solvent followed by centrifugation. The resulting supernatant was injected to LC/MS for the detection of tested peptide. The remaining percentage of each compound was normalized to the 0 hour amount and reported.

## Plasma stability
Peptide was incubated with human plasma at the concentration of 1 mM at 37 C for 1, 2, 3, and 4 hours. The incubation was stopped at indicated time points by addition of organic solvent followed by centrifugation. Parent compound in supernatant was analyzed by LC/MS. Percent remaining of peptide was calculated against the amount of compound at time 0.

**p53 beta-lactamase reporter gene cellular functional assay.**
HCT116 cells were stably transfected with a p53 responsive β-lactamase reporter and expanded in McCoy's 5 A Medium with 10% fetal bovine serum (FBS), Blasticidin, and Penicillin/Streptomycin and then transferred to 1.5 ml freezing vials and stored under liquid nitrogen in growth media containing 5% DMSO. One day prior the assay, a vial of banked cells was recovered in a cell culture flask and incubated for 24 hours, followed by removal of cell growth media and replacement with Opti-MEM containing 2% FBS. The cells were then seeded into a 384-well plate at a density of 8000 cells per well. Peptides were then dispensed to each well using a liquid handler, ECHO 555, and incubated 16 h. The final working concentration of DMSO was 0.5%. β-lactamase activity was detected using the ToxBLAzer Dual Screen (Invitrogen), as per the manufacturer's instructions. Measurements were made using the Envision multiplate reader (Perkin–Elmer). Maximum p53 activity was defined as the amount of β-lactamase activity induced by 50 µM azide-ATSP-7041. This was determined as the highest amount of p53 activity induced by azide-ATSP-7041 from titrations on HCT116 cells. Measurements were taken a minimum of three times (biological replicates) and values are reported as geometric means of the replicates.

**Lactate dehydrogenase (LDH) release assay.** HCT116 cells were stably transfected with a p53 responsive β-lactamase reporter and expanded in McCoy's 5 A Medium with 10% fetal bovine serum (FBS), Blasticidin, and Penicillin/Streptomycin and then transferred to 1.5 ml freezing vials and stored under liquid nitrogen in growth media containing 5% DMSO. One day prior the assay, a vial of banked cells was recovered in a cell culture flask and incubated for 24 hours, followed by removal of cell growth media and replacement with Opti-MEM containing 2% FBS. The cells were then seeded into a 384-well plate at a density of 8000 cells per well. Peptides were then dispensed to each well using a liquid handler, ECHO 555, and incubated 16 h. The final

working concentration of DMSO was 0.5%. Lactate dehydrogenase release was detected using the CytoTox-ONE Homogenous Membrane Integrity Assay Kit (Promega), as per the manufacturer's instructions. Measurements were carried out using the Tecan plate reader. Maximum LDH release was defined as the amount of LDH released as induced by the lytic peptide (iDNA79) and used to normalize the results. Measurements were taken a minimum of three times (biological replicates) and values are reported as geometric means of the replicates.

**Tetracycline beta-lactamase reporter gene cellular assay (counterscreen).** This assay was based on Jump-In™ T-REx™ CHO-K1 BLA cells containing a stably integrated β-lactamase under the control of an inducible cytomegalovirus (CMV) promoter. Cells were maintained in Dulbecco's Minimal Eagle Medium (DMEM) with 10% fetal bovine serum (FBS), Blasticidin, and Penicillin/Streptomycin and then transferred to 1.5 ml freezing vials and stored under liquid nitrogen in growth media containing 5% DMSO. One day prior the assay, a vial of banked cells was recovered in a cell culture flask and incubated for 24 hours, followed by removal of cell growth media and replacement with Opti-MEM containing 2% FBS. Cells were seeded into a 384-well plate at a density of 4000 cells per well. Peptides were then dispensed to each well using a liquid handler, ECHO 555 and incubated for 16 h. The final working concentration of DMSO was 0.5%. β-lactamase activity was detected using the ToxBLAzer Dual Screen (Invitrogen), as per the manufacturer's instructions. Measurements were carried out using the Envision multiplate reader (Perkin–Elmer). Counterscreen activity was defined as the amount of β-lactamase activity induced by tetracycline. Measurements were taken a minimum of three times (biological replicates) and values are reported as geometric means of the replicates.

**Circular dichroism (CD).** A total of 5 μL of the 10 mM stock peptide was mixed with 45 μL of 100% methanol, and dried for 2 h in the SpeedVac concentrator (Thermo Scientific). The dried peptide was reconstituted in a buffer (1 mM Hepes pH 7.4 and 5% methanol) to a concentration of 1 mM. The peptide sample was placed in a quartz cuvette with a path length of 0.2 cm. The peptide concentration was determined by the absorbance of the peptide at 280 nM. The CD spectrum was recorded from 300 to 190 nm using the Chirascan-plus qCD machine (Applied Photophysics, Surrey, UK), at 25 °C. All experiments were done in duplicates. The CD spectrum was converted to mean residue ellipticity before deconvolution and estimation of the secondary structure components of the peptide using the CDNN software (distributed by Applied Photophysics). Measurements were taken at least twice and were reported as arithmetic means.

**Cell proliferation studies.** Routine CellTiter-Glo®Cell Proliferation Assays: CellSensor™ p53RE-bla HCT-116 Cell Line was obtained from Invitrogen (K1640) and the Ca Ski cell line was obtained from American Type Culture Collection (CRM-CRL-1550). The medium for CellSensor™ p53RE-bla HCT-116 cells was McCoy's 5 A Medium (GIBCO16600-002) supplied with 10% FBS (Hyclone, SH30406.05), 100 U/100ug/ml of Pen/Strep (SolarBio, P1400) and 5 μg/ml of blasticidin (GIBCO, A11139). The medium for Ca Ski cells was RPMI1640 (Hyclone, SH3080901B) supplied with 10% FBS and 100 U/100ug/ml of Pen/Strep. All cells were kept in humidified incubators with 5% CO2 at 37 °C. Culture media was changed after one day of subculturing and cells were passed again when there was 70–80% confluence. One day before the assay, the cells were trypsinized, collected, counted and viability was determined. The cells were seeded into a 384-well plate (Corning, 3570). The seeding density of CellSensor™ p53RE-bla HCT-116 was 750 cells/well/40 uL and the seeding density of Ca Ski was 600 cells/well/40 uL. The plates were kept in humidified incubators with 5% CO2 at 37 °C overnight. Compounds were prepared as a 10 point, 3x serial dilution in 100% DMSO, starting from a top concentration of 10 mM. Compounds were added to the assay plate by an

Echo 550(Labcyte) and incubated with the cells for 72 hours in humidified incubators with 5% CO2 at 37 °C. At the end of the incubation the assay plate was equilibrated at room temperature for approximately 30 minutes prior to the addition of 25 uL of CellTiter-Glo® 2.0 Reagent. The assay plate was mixed for 2 minutes on an orbital shaker to induce cell lysis. Next, the plate was incubated at room temperature for 10 minutes to stabilize the luminescent signal prior to recording the luminescence on a plate reader.

Extend CellTiter-Glo® Cell Proliferation Panel: Cell proliferation inhibition of compounds were performed on extended cell panels at fee-for-service at Shanghai ChemPartner Co., Ltd. Cell lines were procured form ATCC and included HCT116 (CCL-247), Ca Ski (CRL-1550), SJSA-1 (CRL-2098), A549(CCL-185), Hep G2 (HB-8065), MCF-7 (HTB-22), LoVo (CCL-229), SW48 (CCL-231), SW480(CCL-228), AsPC-1(CRL-1682), HL-60 (CCL-240), LS123 (CCL-255), MDA-MB-361 (HTB-27), MIA PaCa-2 (CRL-1420), HeLa (CCL-2), MS571 (HTB-34), SiHa (HTB-35), NCI-H1299 (CRL-5803), and Saos-2 (HTB-85). Amongst the cell lines used, the following lines are amongst those listed as frequently misidentified by the International Cell Line Authentication Committee: HeLa, A549, HepG2, MCF-7,LoVo, SW48,.AsPC-1,HL-60, and MIA PaCa-2. These latter lines were used in order to employ a variety of lines represented diverse cancer types and that represented different p53 status (wild-type, mutant, null, and HPV infected). Imporantly, assay ready cells were used for these experiments and PCR analysis of target short tandem repeat (STR) markers was used to authenticate cell lines prior to cell banking. Cell viabilities were determined using the CellTiter-Glo® assay kit (Promega G7558). Cells were cultured and seeded onto 384-well assay plate (Corning 3765) using the Thermo Scientific Multidrop. Assay plates were incubated at 37 °C, 5% CO2 overnight to allow cell attachment. Following day, compounds were dispensed into the assay plate using the Tecan HPD300 system and incubated at 37 °C, 5% CO2 for 120 h. At 120 h post dose, CellTiter-Glo assay was performed according to the manufacturer's protocol. Assay plates were read on a Envision Reader with luminescence at an integration time of 0.1 second per well. Staurosporine (Selleckchem S1421) was used as a positive control for the assay.

**Nanoclick permeability assay.** This assay was previously described in detail[22]. The assay was performed in 384-well white assay plates (PerkinElmer CUSG03874) using assay-ready frozen cells that had been transiently transfected with the NanoLuc-HaloTag vector that were thawed at plated in complete media at a density of 6000 cells/ well and incubated at 37 °C 5% CO2 overnight. DIBAC-CA was diluted in assay buffer (OptiMem without phenol red + 1% FBS) and added to cells at a final concentration of 3 μM and incubated at 37 °C 5% CO2 for 1 h. Cells were subsequently centrifuged using a BlueWasher instrument (Blue Cat Bio) to remove the DIBAC-CA solution and washed two times with HBSS (Ca2 + , Mg2 + ). A volume of 30 μL of assay buffer was added back to cell plates. Peptides were serially diluted 4-fold in DMSO with a Hamilton Star apparatus and then delivered into assay plates with an acoustic liquid handler Labcyte ECHO (300 nl, 1% DMSO in-well concentration). After incubating cells with peptide for the desired time (4 or 18 h), the HaloTag ligand, NanoBret618-azide, was added to each well at a final concentration of 10 μM. After 1 h incubation at 37 °C, 5% CO2, a solution of NanoBRET Nano-Glo Substrate and extracellular NanoLuc Inhibitor was diluted in OptiMem to yield a final in-well concentration of 1× Nano-Glo Substrate and 20 μM of NanoLuc inhibitor and read on the Envision instrument immediately. Following addition of NanoBRET Nano-Glo Substrate, donor emission (450 nm) and acceptor emission (610 nm) were measured using an Envision instrument (PerkinElmer).

## In vivo xenograft models

Human osteosarcoma SJSA-1 cells were purchased from ATCC (Manassas, VA, USA) and cultured with RPMI 1640 medium (Sigma Aldrich,

Merck, Germany), supplemented with 10% fetal bovine serum (FBS), 1% l-glutamine, and 1% penicillin/streptomycin. Cells were cultured for less than 3 months after purchase.

Female immunocompromised BALB/c nu/nu mice ($N = 32$, age = 5–8 weeks) were housed in individual ventilated cages and maintained in a 12-hour light-dark cycle (6 am to 6 pm), at a temperature of 19–24 °C with relative humidity of 45–58% and fed ad libitum. 2.5 ×106 SJSA-1 cells were injected subcutaneously in the right lower flank using a 1:1 mix of matrigel and PBS. Tumor size was measured using a digimatic caliper (Mitutoyo, Sweden) and volume was calculated as 4πabc/3 where a, b, and c were measured diameters in all dimensions.

For the comparison of the efficacy and mechanism of action of the drugs used in the present study, we do not expect any difference based on the sex of the animals. However, we are aware that some xenografts may grow at different rates in males and females. The choice of females in this study serves to minimize variability (reduce biological variation) and increase the quality of the study while respecting the three R's by not extending it to double the number of animals where males are also used.

Animals were randomized into four treatment groups with 8 animals/group. Peptides were dissolved in DMSO at 30 mg/ml and administered i.v. in 30% (2-Hydroxypropyl)-β-cyclodextrin (HPβCD) or Sulfobutylether-β-Cyclodextrin (SBEβCD) (Sigma Aldrich, Merck, Germany). When tumors reached a size of 60–100 mm$^3$, mice received at total of seven doses of ATSP-7041, ALRN-6924, MP-616 (alkyne staple) or vehicle twice a week with 30 mg/kg for twenty-five days. The vehicle group was treated with the same volume of DMSO and HPβCD. Tumor measurements were performed in connection with injections for the treatment (not blinded). Mouse weight and tumor growth were monitored every other day. All experiments complied with Swedish law and were performed with permission from the Uppsala Committee of Animal Research Ethics, permit #10966/2020.

## Peptide synthesis

All peptides were sourced from CPC Scientific or made in-house at A*STAR. The purity and identity of the peptides was confirmed by analytic HPLC and mass spectrometry. All the final peptides have ≥90% purity. All peptides were dissolved in neat DMSO as 10 mM stock solution and diluted thereof for subsequent experiments. HATU is hexafluorophosphate azabenzotriazole tetramethyl uronium, HBTU is 2-(1H-benzotriazole-1-yl)-1,1,3,3-tetramethyluronium hexafluorophosphate, DIC is N,N'-diisopropylcarbodiimide, HOBt is 1-hydroxybenzotriazole, NMM is N-methylmorpholine, PyAOP is (7-azabenzotriazol-1-yloxy)tripyrrolidinophosphonium hexafluorophosphate, RCM is ring closing metathesis.

Peptides were synthesized using Knorr Amide MBHA Resin and Fmoc-protected amino acids, and coupled sequentially with HATU/NMM, HBTU/NMM, or DIC/HOBt, activating agents, depending on the peptide and residue being coupled. Amino acid (AA) and couple agent ratios were as follows: AA/HATU/NMM = 3/2.85/6, AA/HBTU/NMM = 3/2.85/6, and AA/HOBT/DIC = 3/3/3. The resin was first swelled in dimethylformamide (DMF) for 2.0 hours followed by addition of 20% piperidine in DMF. The mixture was kept at room temperature for 0.5 hour while a stream of nitrogen was bubbled through it. The mixture was filtered, and the peptidyl resin was washed five times with DMF. For couplings, the Fmoc-protected amino acid, DMF, NMM and HBTU or HATU were added into the resin sequentially. The suspension was kept at room temperature for 1 hour while a stream of nitrogen was bubbled through it. After the ninhydrin test had indicated complete coupling, the mixture was filtered, and the peptidyl resin was washed three times with DMF. 20% Piperidine in DMF was added into the resin to remove the Fmoc group. Subsequent amino acids were coupled to the resin bound peptide sequentially with reaction times between 0.5 and 1.5 hours, depending on the residues being coupled. After assembly of the linear peptide, the resin was washed twice with

MeOH, twice with DCM, and twice and twice with MeOH. The resin was dried under vacuum overnight.

For peptides with olefin staples, RCM was performed on-resin. The peptidyl resin was washed twice with 1,2-dichloroethane (DCE), followed by the addition of Grubbs(I) Catalyst dissolved in DCE (e.g. 150 mg of catalyst in 6 ml of DCE). The mixture was kept at 35 °C for 3 hours while a stream of nitrogen was bubbled through it. The mixture was filtered, and the peptidyl resin was washed three times with DMF (e.g., 3 × 20 mL). After the RCM was complete, a test cleavage was performed to ensure adequate yield.

For peptides with lactam staples, Fmoc-Lys(Mtt)-OH was used and selectively deprotected by washing the resin twice with DCM, followed by addition of a 2%TFA/4%TIS/94%DCM solution. The mixture was kept at room temperature for 5 min while a stream of nitrogen was bubbled through it. The resin was then washed with DCM twice. This procedure was repeated 5 times before proceeding. 5 mL DMF, HOBT (10eq), DIPEA (10eq) and PyAOP (3eq) were added into the resin sequentially. The mixture was shaken at room temperature overnight and mass spectrometry had indicated a complete reaction. The resin was then washed with DMF twice.

For peptide containing the dialkyne staple, stapling was performed on resin. The peptidyl resin was dried by washing three times with DCM, and then transferred to a glass vial. THF (10 mL), DIPEA (5 mL), Pd(PPh$_3$)$_2$Cl$_2$ (0.05 mmol 35 mg) and Cu(I) (0.1 mmol 20 mg) were and the mixture was put on a table concentrator rotated at -30 °C under air for 16 hours. A test cleavage was then performed to ensure adequate yield and the solution was transferred to the reactor and filtered, and then washed six times with DMF.

For peptides with acetylated N-termini, an acetic anhydride/NMM capping solution was applied for 30 minutes after the final coupling reaction. Peptides were cleaved and then purified as a mixture of cis-trans isomers by RP-HPLC.

## Computational methods

Peptide design and molecular dynamics simulations were carried out using the AMBER18 package[47] following a protocol used earlier[30] and are discussed below. The models of the peptides used in this study were based on the structural models of Mdm2: ATSP-7041, Mdm2: $^d$PMI-δ (6-10 staple) and Mdm2: $^d$PMI-δ (5-12 staple) published earlier[6,30].

All the structural models built were subjected to Molecular Dynamics (MD) simulations for further refinement. MD simulations were carried out on the free peptides and peptide – Mdm2 complexes. The Xleap module of AMBER18[48] was used to prepare the system for the MD simulations. Hydrogen atoms were added and the N- and C-termini of the peptides were capped with ACE and NHE moieties respectively. The parameters for the staple linkers were taken from our previous studies[6,30]. All the simulation systems were neutralized with appropriate numbers of counter ions and solvated in an octahedral box with TIP3P[49] water molecules, leaving at least 10 Å between the solute atoms and the borders of the box. MD simulations were carried out with the pmemded module of the AMBER 18 package in combination with the ff14SB force field[50]. We have found that the combination of ff14SB and TIP3P and the parameters listed below have been successful in simulating the Mdm2-peptide systems[50,51]. All MD simulations were carried out in explicit solvent at 300 K. During the simulations, the long-range electrostatic interactions were treated with the particle mesh Ewald[51] method using a real space cut off distance of 9 Å. The settle[52] algorithm was used to constrain bond vibrations involving hydrogen atoms, which allowed a time step of 2 fs during the simulations. Solvent molecules and counter ions were initially relaxed using energy minimization with restraints on the protein and peptide atoms. This was followed by unrestrained energy minimization to remove any steric clashes. Subsequently the system was gradually heated from 0 to 300 K using MD simulations with positional restraints (force constant: 50 kcal mol$^{-1}$ Å$^{-2}$) on protein and peptides over a period of 0.25 ns

allowing water molecules and ions to move freely followed by gradual removal of the positional restraints and a 2 ns unrestrained equilibration at 300 K. The resulting systems were used as starting structures for the respective production phases of the MD simulations. For each case, three independent (using different initial random velocities) MD simulations were carried out starting from the well equilibrated structures. Each MD simulation was carried out for 250 ns and conformations were recorded every 4 ps. To enhance the conformational sampling, each peptide was subjected to Biasing Potential Replica Exchange MD (BP-REMD) simulations[53]; the BP-REMD technique is a type of Hamiltonian-REMD method but includes an additional biasing potential to promote dihedral transitions along the replicas[53]. For each system, BP-REMD was carried out with eight replicas including a reference replica without any bias. BP-REMD was carried out for 50 ns with exchange between the neighbouring replicas attempted every 2 ps and accepted or rejected according to the metropolis criteria[54]. Conformations sampled at the reference replica (no bias) were used for further analysis. Simulation trajectories were visualized using VMD[55] and figures were generated using Pymol[56].

### Reporting summary
Further information on research design is available in the Nature Portfolio Reporting Summary linked to this article.

## Data availability
The full processed data set generated in this study is available as Supplementary Data. Raw data is available by request from the corresponding author Brian Henry at MSD Singapore.

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

## Acknowledgements

We thank Evans (Chen) Ge, Mike (Dixin) Xue, and Simon (Junhua) Li at Chinese Peptide Company (CPC) for peptide synthesis support. We thank Xin Dong, Jingxi Zhang, and Jie Wang at Pharmaron for biochemical and cellular assay support. Funding support from the Agency for Science, Technology and Research (A*STAR), Singapore Industry Alignment Fund-Pre-Positioning (IAF-PP) grant (H1701a0010) and Singapore Industry Alignment Fund-Industry Collaboration Project (IAF-ICP) grant (I1901E0039) were gratefully acknowledged.

## Author contributions

Conceptualization, A.C., H.J., J.H., B.H., C.S.V., T.K.S., D.P.L., R.G., S.K., C.J.B., C.W.J., A.W.P.; methodology, A.C., H.J., T.Y.Y., S.K., C.J.B., A.W.P.; investigation, A.C., H.J., T.Y.Y., R.D., D.S., L.Y., Y.C.A.J., L.G., P.A., H.Y.K.K., Y.H.L., S.K., C.J.B., A.W.P., formal analysis, A.C., H.J., L.Y., B.S., J.H., S.K., C.J.B., A.W.P.; writing—original draft, A.C., R.D., JH., T.KS., S.K., C.J.B., C.W.J., A.W.P. Writing—review and editing, A.C., C.W.J., A.W.P. Resources, K.B., B.H., M.N., C.S.V., T.KS., D.P.L., C.J.B., C.W.J., A.W.P. Project administration, A.C., H.J., Y.H.L., B.H., C.S.V., T.K.S., D.P.L., R.G., S.K., C.J.B., C.W.J., A.W.P.; supervision, Y.H.L., A.P., J.H., S.L., B.H., M.N., C.S.V., T.K.S., D.P.L., R.G., S.K., C.J.B., C.W.J., A.W.P.

## Competing interests

The authors declare no competing interests.
