## [Peer Review File · Nature Communications]

REVIEWER COMMENTS

Reviewer #1 (Remarks to the Author):

The manuscript focuses on the determination of a series of maxims for the development of stapled peptides. The authors have chosen as their model system the well-developed p53-MDM2 target, since a series of stapled peptides mimicking the p53 binding helix are known and cellular assays for the cell penetration and binding of these peptides are well developed. The manuscript describes experiments to probe the effects of changes to the lipophilicity and helicity of a stapled peptide lead on its cell penetration and activity; then tests initial design strategies against other stapled peptide leads targeting the same interaction.

The manuscript is underpinned by a series of well-established assays and modelling techniques. The authors are experts in this field, particularly relating to this therapeutic target and are well-placed to conduct this work.

Whilst the study is undoubtedly thorough, with data reported for over 350 individual stapled peptide sequences, and covering (albeit briefly) different staple methodologies, the authors acknowledge themselves the major limitation that the work is directed towards one therapeutic target. Thus, the general framework presented might not be as general as suggested.

The data and methodology used in the study appear appropriate; the use of cellular assays such as the Nanoclick assay to assess the cellular permeability of peptides directly (even when not bound to the target) using a fluorescent readout is particularly current.

The narrative of the paper is clear, but the literature cited in the manuscript is not as broad-based as it could be and does not reflect the field as well as it could. There are several recent reviews and perspective articles on approaches to the development of stapled peptides, which also give insight the design of stapled peptides and yet are not referenced. Similarly, previous comprehensive analysis of cellular uptake across a range of stapled peptides by the Verdine and Walensky groups has not been cited.

Overall, this is an excellent study, but not one which is appropriate for publication in Nature Communications. Given the target-specific nature of the data presented, it might appeal better to a more focused audience such as the ACS Publications Journal of Medicinal Chemistry.

Reviewer #2 (Remarks to the Author):

This is an outstanding contribution to the field of stapled peptides and PPIs, and particularly for targeting mdm2/mdmx and it is very insightful for this target. I have couple of suggestions/comments:

1) the authors suggest that the rules identified for this model stapled-peptide/target can be generalised for stapled peptides and propose a workflow to design lead stapled peptides against an intracellular target. These rules seem to be specific for helical amphipathic stapled peptides and for this target and might not apply for all intracellular targets and peptide-inhibitors. For instance, this helical peptide and all the residues involved in binding are hydrophobic, this might not be the case for other peptide/target pairs; not all the PPIs are inhibited by helical peptides; not all peptide binders identified from library screening (step 1) will be cell permeable, even after being chemically stapled; removing all the positive charges while leaving hydrophobic residues and including chemical staples might still be toxic via membrane-disruption. The opposite approach (i.e. removing hydrophobic residues while keeping positive charges) can also work to keep the activity and cell penetrating properties for some peptides.

2) This is a very data dense manuscript, and in places it requires an expert knowledge in peptide chemistry to be understood. To make it clear for the broader scientific audience of Nat Comms some concepts might need some explanation.

Reviewer #3 (Remarks to the Author):

The manuscript by Chandramohan et al is an important contribution to the field of p53 activation, and more broadly, the field of stabilized peptides. The sheer volume of analogs made and analyzed via a workflow that outputs relevant, compelling, and high-quality data is impressive and laudable. The authors mined the data to draw sound conclusions and vetted those ideas on a related yet divergent peptide, which is to be commended. However, this reviewer was left with an incomplete feeling as the style and presentation of the results made it feel like there was more to be learned and gleaned as the figures generally were presented in a descriptive fashion as opposed to synthesizing a data set. Specifically, the helical wheel plots are not intuitive. For 2a & 2b, what position tolerates polar residues best or charged residues? That info is in the figure but it could be highlighted to a greater extent. Consider simply putting the letters in a quadrant based on high or low hydrophobicity/polar/neg/pos, yet maintaining the colors so at a glance it would reveal the preferences for each position.

2c is particularly vexing as the binding and cellular were presented already in greater clarity in 2a & 2b, so including it here only complicated things. Why not just present the nano click and solubility with added granularity as in 2a/b.

2d seems capable of conveying additional information by coloring the dots based on a additional chosen relevant parameter.

Also, what is the cell ratio? The authors need to define and convey this correlation with a demonstrative figure. Similarly, what is the permeability ratio. These seem like useful properties for the field so if they are introduced here for the first time, please elaborate on them, and illustrate the relevance with a few exemplary peptides.

For Table 1 & 2, sort by a relevant column and then color cells that agree or break with a trend or are relevant for conclusions drawn in the text.

Figure 7 is borderline uninterpretable.

The main frustration is that, in this reviewer's opinion, the results are described in a far clearer manner with words in the text than with the figures. Otherwise, it's a great paper and my suggestions are only to make it more readable and intuitive for the reader.

5 August, 2023

Response to comments from the Reviewers:

Reviewer #1 (Remarks to the Author):

The manuscript focuses on the determination of a series of maxims for the development of stapled peptides. The authors have chosen as their model system the well-developed p53-MDM2 target, since a series of stapled peptides mimicking the p53 binding helix are known and cellular assays for the cell penetration and binding of these peptides are well developed. The manuscript describes experiments to probe the effects of changes to the lipophilicity and helicity of a stapled peptide lead on its cell penetration and activity; then tests initial design strategies against other stapled peptide leads targeting the same interaction.

The manuscript is underpinned by a series of well-established assays and modelling techniques. The authors are experts in this field, particularly relating to this therapeutic target and are well-placed to conduct this work.

Whilst the study is undoubtedly thorough, with data reported for over 350 individual stapled peptide sequences, and covering (albeit briefly) different staple methodologies, the authors acknowledge themselves the major limitation that the work is directed towards one therapeutic target. Thus, the general framework presented might not be as general as suggested.

- *Thank you to Reviewer 1 for highlighting this concern. As noted, we had already highlighted this ourselves in the discussion. However, we concur that this point could be expanded upon and made clearer in other parts of the text. In accordance, we have toned down the claims in the following places (highlighted in yellow in the resubmitted version):*

1. *Abstract: we highlighted that although the work was translatable across two distinct series, that both were against the same target. We also rewrote the final sentence of the abstract to suggest that our findings are likely to be most applicable to targets that share characteristics with Mdm2/X.*
2. *We restate this latter point in the discussion and then expound upon it in a manner that leverages the insightful comments from reviewer 2 (see below).*

The data and methodology used in the study appear appropriate; the use of cellular assays such as the Nanoclick assay to assess the cellular permeability of peptides directly (even when not bound to the target) using a fluorescent readout is particularly current.

The narrative of the paper is clear, but the literature cited in the manuscript is not as broad-based as it could be and does not reflect the field as well as it could. There are several recent reviews and perspective articles on approaches to the development of stapled peptides, which also give insight the design of stapled peptides and yet are not referenced. Similarly, previous comprehensive analysis of cellular uptake across a range of stapled peptides by the Verdine and Walensky groups has not been cited.

- *Highlighting this issue is appreciated. We have rectified the oversight and have added a number of key citations, including several from both the Verdine and Walensky groups. We have also now included a*

relevant paper from Aileron Therapeutics that has come out in the interim. Here is the list of newly cited work:

Nat Protoc 2011 Jun;6(6):761-71. doi: 10.1038/nprot.2011.324. Epub 2011 May 12.
Synthesis of all-hydrocarbon stapled α -helical peptides by ring-closing olefin metathesis
Young-Woo Kim 1, Tom N Grossmann, Gregory L Verdine

De novo mapping of α -helix recognition sites on protein surfaces using unbiased libraries.
Li K, Tokareva OS, Thomson TM, Wahl SCT, Travaline TL, Ramirez JD, Choudary SK, Agarwal S, Walkup WG 4th, Olsen TJ, Brennan MJ, Verdine GL, McGee JH.

J Am Chem Soc 2007 Mar 7;129(9):2456-7. doi: 10.1021/ja0693587. Epub 2007 Feb 7.
Reactivation of the p53 tumor suppressor pathway by a stapled p53 peptide
Federico Bernal 1, Andrew F Tyler, Stanley J Korsmeyer, Loren D Walensky, Gregory L Verdine

Nat Chem Biol
. 2016 Oct;12(10):845-52. doi: 10.1038/nchembio.2153. Epub 2016 Aug 22.
Biophysical determinants for cellular uptake of hydrocarbon-stapled peptide helices
Gregory H Bird 1, Emanuele Mazzola 2, Kwadwo Opoku-Nsiah 1, Margaret A Lammert 1, Marina Godes 1, Donna S Neuberg 2, Loren D Walensky 1

Oncogene. 2017 Apr; 36(15): 2184–2190.
Mechanistic Validation of a Clinical Lead Stapled Peptide that Reactivates p53 by Dual HDM2 and HDMX Targeting
Franziska Wachter,1 Ann M. Morgan,1 Marina Godes,1 Rida Mourtada,1,2 Gregory H. Bird,1 and Loren D. Walensky1,*

Proc Natl Acad Sci U S A
. 2010 Aug 10;107(32):14093-8. doi: 10.1073/pnas.1002713107. Epub 2010 Jul 21.
Hydrocarbon double-stapling remedies the proteolytic instability of a lengthy peptide therapeutic
Gregory H Bird 1, Navid Madani, Alisa F Perry, Amy M Princiotto, Jeffrey G Supko, Xiaoying He, Evripidis Gavathiotis, Joseph G Sodroski, Loren D Walensky

ACS Chem Biol
. 2015 Jun 19;10(6):1362-75. doi: 10.1021/cb501020r. Epub 2015 Mar 31.
Hydrocarbon stapled peptides as modulators of biological function
Philipp M Cromm 1 2, Jochen Spiegel 1 2, Tom N Grossmann 1 2 3

Guerlavais V, Sawyer TK, Carvajal L, Chang YS, Graves B, Ren JG, Sutton D, Olson KA, Packman K, Darlak K, Elkin C, Feyfant E, Kesavan K, Gangurde P, Vassilev LT, Nash HM, Vukovic V, Aivado M, Annis DA.
Discovery of Sulanemadlin (ALRN-6924), the First Cell-Permeating, Stabilized α -Helical Peptide in Clinical Development. *J Med Chem*. 2023 Jul 13. doi: 10.1021/acs.jmedchem.3c00623. Epub ahead of print. PMID: 37439511.

Overall, this is an excellent study, but not one which is appropriate for publication in Nature Communications. Given the target-specific nature of the data presented, it might appeal better to a more focused audience such as the ACS Publications Journal of Medicinal Chemistry.

Reviewer #2 (Remarks to the Author):

This is an outstanding contribution to the field of stapled peptides and PPIs, and particularly for targeting mdm2/mdmx and it is very insightful for this target. I have couple of suggestions/comments: 1) the authors suggest that the rules identified for this model stapled-peptide/target can be generalised for stapled peptides and propose a workflow to design lead stapled peptides against an intracellular target. These rules seem to be specific for helical amphipathic stapled peptides and for this target and might not apply for all intracellular targets and peptide-inhibitors. For instance, this helical peptide and all the residues involved in binding are hydrophobic, this might not be the case for other peptide/target pairs; not all the PPIs are inhibited by helical peptides; not all peptide binders identified from library screening (step 1) will be cell permeable, even after being chemically stapled; removing all the positive charges while leaving hydrophobic residues and including chemical staples might still be toxic via membrane-disruption. The opposite approach (i.e. removing hydrophobic residues while keeping positive charges) can also work to keep the activity and cell penetrating properties for some peptides.

- *As noted above, we have toned down our remarks to acknowledge more clearly that the insights gained here might be most applicable to Mdm2/X and targets with similar characteristics. In addition, we'd like to thank this reviewer in particular as their comments helped us craft a more thoughtful discussion around this point. Indeed, we have incorporated many of the remarks outlined above in the discussion.*

2) This is a very data dense manuscript, and in places it requires an expert knowledge in peptide chemistry to be understood. To make it clear for the broader scientific audience of Nat Comms some concepts might need some explanation.

- *We appreciate the point and have added some clarifying remarks – not only in terms of peptide chemistry but also to assays that are particular to the field. **These additions are highlighted in turquoise in the resubmitted version.***

Reviewer #3 (Remarks to the Author):

The manuscript by Chandramohan et al is an important contribution to the field of p53 activation, and more broadly, the field of stabilized peptides. The sheer volume of analogs made and analyzed via a workflow that outputs relevant, compelling, and high-quality data is impressive and laudable. The authors mined the data to draw sound conclusions and vetted those ideas on a related yet divergent peptide, which is to be commended. However, this reviewer was left with an incomplete feeling as the style and presentation of the results made it feel like there was more to be learned and gleaned as the figures generally were presented in a descriptive fashion as opposed to synthesizing a data set. Specifically, the helical wheel plots are not intuitive. For 2a & 2b, what position tolerates polar residues best or charged residues? That info is in the figure but it could be highlighted to a greater extent. Consider simply putting the letters in a quintant based on high or low hydrophobicity/polar/neg/pos, yet maintaining the colors so at a glance it would reveal the preferences for each position.

- *we greatly appreciate the comments from reviewer 3 as they have helped us craft a manuscript that is much improved in terms of data presentation. This should help better communicate our findings, which we value a great deal. We have improved data presentation in several ways as highlighted here and below for subsequent points.*

- *We are unfamiliar with the term quintant. However, for Figs 2a and 2b, we have done a major revamp to aid viewer data interpretation. Specifically, we have now colored the residues according to polarity*

and charge state. Furthermore, we have added icons to indicate the effect of the substitutions. We feel the figure is much more interpretable now and thank the reviewer for prompting these changes.

2c is particularly vexing as the binding and cellular were presented already in greater clarity in 2a & 2b, so including it here only complicated things. Why not just present the nano click and solubility with added granularity as in 2a/b.

- We have followed reviewer 3's advice and removed the binding and cellular data from 2c to focus on the new data not presented elsewhere. We have also added colored icons beside each position to clearly mark the effect of the glutamic acid substitution on NanoClick permeability. For clarity, we have also removed the solubility values from this figure since all peptides had good solubility. Instead, we have referred readers to Table S2 within the text since those numbers are already displayed there.

2d seems capable of conveying additional information by coloring the dots based on a additional chosen relevant parameter.

- This is a valuable suggestion. We have chosen to highlight peptides according to charge as this better highlights that permeability under the 0% serum condition is enhanced by removing charged residues. We have expounded upon this point in the text.

Also, what is the cell ratio? The authors need to define and convey this correlation with a demonstrative figure. Similarly, what is the permeability ratio. These seem like useful properties for the field so if they are introduced here for the first time, please elaborate on them, and illustrate the relevance with a few exemplary peptides.

- Thank you to Reviewer 3 for catching this omission. As now better explained in the text, cell ratio is the ratio of the cell potency to the biochemical Kd (a metric for determining permeability). Permeability ratio is the same thing. To avoid confusion, we now use cell ratio exclusively in the text.

For Table 1 & 2, sort by a relevant column and then color cells that agree or break with a trend or are relevant for conclusions drawn in the text.

- Peptides are listed both by order they appear in the text and by class. To better orient the viewer, we have now i) added a new column called "Series Class" and ii) color coded the values according to the magnitude of the effect.

Figure 7 is borderline uninterpretable.

- we appreciate the reviewer's frank comment. This is much appreciated as it prompted us to make a new figure that we feel is much clearer. Portions of the previous figure 7 have been moved to the supplementary figures (Fig S5).

The main frustration is that, in this reviewer's opinion, the results are described in a far clearer manner with words in the text than with the figures. Otherwise, it's a great paper and my suggestions are only to make it more readable and intuitive for the reader.

- We sincerely thank Reviewer 3 for their comments as we feel they helped greatly improve the paper.

REVIEWERS' COMMENTS

Reviewer #3 (Remarks to the Author):

Great revision... no further comments.